# Stress-Related Dysfunction of Adult Hippocampal Neurogenesis—An Attempt for Understanding Resilience?

**DOI:** 10.3390/ijms22147339

**Published:** 2021-07-08

**Authors:** Julia Leschik, Beat Lutz, Antonietta Gentile

**Affiliations:** 1Institute of Physiological Chemistry, University Medical Center of the Johannes Gutenberg University Mainz, 55128 Mainz, Germany; beat.lutz@uni-mainz.de; 2Leibniz Institute for Resilience Research (LIR), 55122 Mainz, Germany; 3Synaptic Immunopathology Lab, IRCCS San Raffaele Pisana, 00166 Rome, Italy; gntnnt01@uniroma2.it

**Keywords:** adult neurogenesis, stress, major depressive disorder, resilience

## Abstract

Newborn neurons in the adult hippocampus are regulated by many intrinsic and extrinsic cues. It is well accepted that elevated glucocorticoid levels lead to downregulation of adult neurogenesis, which this review discusses as one reason why psychiatric diseases, such as major depression, develop after long-term stress exposure. In reverse, adult neurogenesis has been suggested to protect against stress-induced major depression, and hence, could serve as a resilience mechanism. In this review, we will summarize current knowledge about the functional relation of adult neurogenesis and stress in health and disease. A special focus will lie on the mechanisms underlying the cascades of events from prolonged high glucocorticoid concentrations to reduced numbers of newborn neurons. In addition to neurotransmitter and neurotrophic factor dysregulation, these mechanisms include immunomodulatory pathways, as well as microbiota changes influencing the gut-brain axis. Finally, we discuss recent findings delineating the role of adult neurogenesis in stress resilience.

## 1. Introduction: Adult Neurogenesis

Adult neurogenesis in the mammalian brain is a continuous lifelong physiological process, which dramatically declines during aging [1]. The main work, so far, elucidating the regulatory mechanisms of adult neural stem cells has been done in rodent animal models, whereas the existence of neurogenesis in the adult human brain is still under debate [2,3,4]. Even if many studies report the existence of adult neurogenesis in humans during the whole lifespan [5,6,7,8,9,10], these findings have been questioned by others, which could detect adult neural stem cells and their progeny only in early childhood [11,12,13]. The discrepancy here might arise from different probe sampling and further technical issues, which are extensively described in recent reviews by Lucassen and colleagues [14,15]. Especially, a direct comparison to rodent models seems difficult as brains from healthy human subjects cannot be processed and analyzed similarly to rodents, since human samples usually arise from postmortem fixed tissue [16].

### 1.1. Adult Hippocampal Neurogenesis

The generation of newly built neurons needs to be tightly controlled under physiological conditions. Control mainly occurs on three different levels, which comprise first the proliferation of adult neural stem cells and/or progenitor cells (NPCs), also maintaining the stem cell pool; second, the neuronal and glial determination and differentiation of NPCs. Lastly, newly built neurons need to survive, mature, and functionally integrate into already existing neuronal circuits, which is the final level of regulation. For a detailed description of adult neural stem cell regulation, see the two recent reviews from Obernier and Alvarez-Buylla (2019) and Denoth-Lippuner and Jessberger (2021) [17,18]. In the adult mammalian brain, two main regions are described where new neurons are continuously generated under physiological conditions. This is, on one hand, the subventricular zone (SVZ) of the lateral ventricles, which gives rise to new GABAergic granular and periglomerular neurons of the olfactory bulb. The second main neurogenic region is located in the adult hippocampus, specifically in the subgranular zone (SGZ) of the dentate gyrus (DG), which serves as an input station into the whole hippocampal formation. After cell division in the SGZ, hippocampal neural stem cells differentiate into postmitotic glutamatergic cells of the DG granule cell layer, a process which takes approximately two months from cell birth until the end of maturation [19].

The vast majority of newly built granule neurons are added to the granule cell layer of the DG throughout life, thereby extending the granule cell layer [20]. This implicates a high level of neuronal plasticity, as the addition of new neurons has possibly the capacity to rewire an existing neuronal circuit. 

### 1.2. Adult Hippocampal Neurogenesis in Stress-Related Behavior

Given that the DG is part of the hippocampus, hippocampal function is modulated by changes in rates of adult neurogenesis. Indeed, in most studies, an increase of adult neurogenesis led to enhanced performance in hippocampal-dependent behavioral tasks, whereas a lack or reduction induced impaired hippocampal-dependent tasks [20,21,22,23,24,25]. The hippocampus is part of the limbic system and can be subdivided into the dorsal and the ventral part, both of which exert differential functions. Whereas, the ventral hippocampus is mainly important for mood control and regulating emotional states, the dorsal hippocampus has been predominantly implicated in cognitive functions, such as learning and memory [26,27,28]. Nonetheless, recent studies tend to show that rather than strictly containing dissociated roles, both the dorsal and the ventral hippocampus contribute to the integration of contextual information and context-specific events in a complementary way [29,30]. The hippocampus is regarded as the key brain area involved in regulating stress response [31]. Therefore, proper stress coping is associated with hippocampal processing of emotional and cognitive information. Appropriate stress coping of an individual is important to adopt or to stay in a resilient state as a protective mechanism against symptoms of stress-related psychiatric disorders, such as major depressive disorder (MDD), anxiety disorders, as well as posttraumatic stress disorder (PTSD) [32]. Particularly, adult-born DG granule cells are essential for hippocampal-dependent tasks involving pattern separation, cognitive flexibility, and memory interference, as well as forgetting [33,34,35,36]. All these processes may be relevant for the acquisition of stress resilient outcomes, and their failure could result in stress-related mental dysfunctions. Experiments ablating or reducing adult neurogenesis have demonstrated, besides a lack of spatial memory, the occurrence of depression- and anxiety-like behavior, which, however, in several studies is only detectable in response to stress [20,25,36,37,38,39,40]. Certainly, increasing neurogenesis is sufficient to reduce anxiety and depression-like behaviors [41] and hypothalamus pituitary adrenal (HPA) axis dysregulation [42]. In addition, the finding that rodents showing depressive-like behavior and depressed human individuals display a thinner granule cell layer, whereby antidepressant-treatment restore adult neurogenesis to physiological levels [36,43], suggests adult neurogenesis as a resilience mechanism [44]. In fact, Anacker et al. (2018) [45] recently demonstrated that young adult-born DG granule cells are necessary to confer stress resilience by inhibiting ventral mature granule neurons during chronic social defeat stress (CSDS). In line with this, a direct causal relationship between newborn neuronal activity and affective behavior was demonstrated by Tunc-Ozcan et al. (2019) [46]. The authors reported that activating newborn neurons alleviated depressive-like behavior and reversed the effects of chronic unpredictable stress (CUS). The results further suggest that the mere numbers of newborn neurons are a relatively coarse read-out, but also their neuronal activity and degree of functional integration into the existing neuronal network of the mature DG is a crucial factor in governing resilience. Modulation of network activity particularly applies to young adult-born DG granule cells in the age of 4–6 weeks after cell birth. At four weeks, young newborn DG neurons start to enter a critical period of development with distinct electrophysiological properties, including high input resistance and a lack of GABAergic inhibition, which results in a greater propensity for hyperexcitability and a lower activation threshold than mature DG cells. Furthermore, an enhanced plasticity and long-term potentiation (LTP) is detectable [47,48]. 

For this reason, it is obvious that young adult-born DG granule cells make a unique contribution to hippocampus-dependent behaviors, e.g., novelty-evoked exploration and contextual fear conditioning [49]. Particularly, pattern separation, the ability to transform similar experiences into distinct non-overlapping representations of memory, is thought to be modulated by young adult-born granule neurons [50,51], and it has been shown that increasing neurogenesis in the adult hippocampus is sufficient to improve pattern separation [52]. Interestingly, recent data in rats suggest that differently aged populations of adult-born neurons are implicated in distinct phases of memory formation processing. By using retroviral and chemogenetic approaches, Lods et al. (2021) demonstrated that mature (6-week-old) and immature (1–2-week-old) adult-born neurons are both activated by remote memory retrieval, but that the process of remote memory reconsolidation solely depends on adult-born neurons, which were immature during learning [53]. These findings highlight the importance of adult neurogenesis in established reactivated memories.

In relation to stress resilience, one could speculate that individual differences in neurogenesis-induced memory processing and/or pattern separation could lead to an individual resilient behavior towards stress, e.g., discriminating harmful from harmless situations/contexts. 

Nevertheless, addressing the functional relevance of neurogenesis in stress resilience, most of the studies present correlative results, meaning that changing neurogenesis before analyzing the outcome led to behavioral changes. What is mostly missing, but interesting in this aspect, are studies in which individuals that are resilient demonstrate an “increased” adult neurogenesis per se. Indeed, recent findings in genetically identical mice hint towards individual neurogenesis regulating individual behavioral traits [54].

## 2. Major Depressive Disorder and Adult Neurogenesis

With increasing incidence and a high lifetime prevalence of 10–20% in the human population, MDD is one of the most studied psychiatric diseases [55]. MDD impacts mood and behavior, as well as various physical functions, such as appetite and sleep, and can lead to suicidal behavior. The causes for the development of the disease are multifactorial and not yet completely understood at the neurophysiological and molecular levels. Neuroendocrinological data hint towards a dysregulation of the HPA axis, since patients with hypercortisolism or exogenous glucocorticoid (GC) treatment more often develop MDD than healthy individuals [56]. Furthermore, the GC cortisol in humans and corticosterone in rodents are the most important stress hormones, highly elevated during periods of chronic stress and regarded as the main effector for the development of depression [31]. There is a variety of animal rodent models to mimic MDD symptoms, which basically consist of different stressors applied with distinct timing. In addition, also chronic corticosterone treatment induces a depressive-like phenotype in rodents. For an overview of animal stress models and depressive-like symptoms, see Table 1, and for further description of animal model protocols, a recent review [57]. 

It is commonly known that the hippocampus is an important mediator of the negative feedback of the HPA axis involved in proper stress response [58]. Past studies, using postmortem analysis or magnetic resonance imaging (MRI), have revealed reductions in hippocampal volume of depressed patients [59,60]. Interestingly, in PTSD, a recent study reported a smaller human DG volume pretrauma as a predisposing vulnerability factor [61], which could also apply to MDD.

Like humans, rodents do not all develop depressive-like symptoms after chronic stress exposure, and hence, can be subdivided into resilient and susceptible groups based on their individual behavioral responses to stress [62]. Interestingly, this variation of the stress response can be linked to a reduction of hippocampal volume after CSDS in susceptible compared to non-stressed control mice [63]. Reductions of hippocampal volume could be either due to reduced neuroplasticity by dendritic growth arrest or atrophy leading to shortening of dendritic length and consequently to a reduction in spine density, which was observed in the CA3 region, and/or by the decreased generation of new neurons in the DG [64,65,66]. It is also unknown whether changes in adult neurogenesis and CA3 dendritic morphology are linked or are independent of each other, whereby one study in mice suggests that inhibiting adult neurogenesis for several months can lead to CA3 atrophy [67]. 

In rodent animal models, it is well established that protocols of chronic stress or chronic corticosterone treatment, used as a model of HPA axis overactivity, decrease adult neurogenesis (Table 1). Most, but not all, studies demonstrated deficits in neural stem/progenitor proliferation and/or differentiation, addressing also decreased cellular survival in the SGZ of the adult hippocampus (reviewed and discussed in Levone et al. (2015) [44]). Recent studies suggest that this might also be true for humans, by observing decreased numbers of granule cells in the DG of non-medicated depressed patients compared to healthy individuals and increased hippocampal neurogenesis and granule cell layer volume in antidepressant-treated compared to non-medicated patients [68,69,70]. In humans, early life adversity is one of the risk factors to develop MDD, including suicidal behavior in adulthood [71]. Interestingly, Boldrini et al. (2019) also demonstrated that an increased volume of DG is associated with resilience to early life adversity, presumably due to increased neurogenesis during childhood [72]. 

### 2.1. Antidepressants Acting on Adult Neurogenesis

It is most widely accepted that MDD patients display monoaminergic deficits [109,110], which are restored by treatment with the most common antidepressants targeting the serotonergic and norepinephrinergic systems. The majority of antidepressants need to be administered for at least six weeks to two months until full effectiveness, which opposes the impact of acute functioning. Rather a neuroplasticity-related mechanism is suggested, which seems to involve upregulation of brain-derived neurotrophic factor (BDNF) and thereby the antidepressant-induced enhancement of neurogenesis (see Section 3.1.3) [43,111]. As mentioned above, the full maturation of newly built hippocampal neurons takes approximately two months, and indeed, adult neural stem and precursor cells are positively regulated by serotonin (5-HT) [112] and norepinephrine [113,114,115]. In line with this, ablation studies with X-irradiation or cytostatic agents demonstrated that adult neurogenesis is necessary to ameliorate anxiety- and/or depressive-like behavioral effects exerted by antidepressants [116,117]. Moreover, a recent publication reports that selectively suppressing the excitability of newborn neurons by chemogenetic approaches without changing neurogenesis rate abolishes the antidepressant effect of the selective serotonin reuptake inhibitor (SSRI) fluoxetine, and that remarkably, activation of these neurons is sufficient to alleviate anxiety- and depressive-like behavior [46]. Not necessarily contrasting to this, other studies also demonstrated neurogenesis-independent mechanisms of antidepressants with a pivotal role in inducing remodeling of dendrites and synapses in mood-regulating limbic brain regions, which seems to account for an additional short-term effect of antidepressants [118,119,120].

Interestingly, a recent publication showed that blockade of indolamine 2, 3-dioxygenase 1 (IDO-1), an enzyme of the kynurenine pathway, associated with reduced 5-HT levels and hyperactivated in depression, ameliorated impaired hippocampal neurogenesis and depressive-like symptoms in mice, which underlines the importance of neurogenesis in the mechanistic action of monoamine-increasing antidepressants [111]. It is well known that approximately 30–40% of depressed patients are treatment-resistant by monotherapy with common antidepressants and do not achieve full remission of symptoms, even if medicated with an additional antidepressant after monotherapy [121]. Recent studies have shown that ketamine, an open channel blocker of the N-methyl-D-aspartate receptor (NMDAR), is effective for patients with treatment-resistant depression. Interestingly, similarly to monoaminergic antidepressants, also ketamine seems to act via augmented BDNF expression and a subsequent increase of adult neurogenesis, which was evident in the ventral hippocampus of adult mice [122]. In addition, electroconvulsive therapy (ECT), an efficient treatment for severe and refractory unipolar and bipolar depression, has remarkable antidepressant [123] and proneurogenic [124] properties. The subfield analysis of MRI scans showed that ECT in depressed patients increases the volume of major hippocampal regions and the DG [125,126]. Furthermore, the longitudinal analysis of hippocampal volume showed that hippocampal baseline is predictive of subsequent clinical outcomes [127]. Of note, the latter finding that is suggestive of increased neurogenesis is corroborated by studies with electroconvulsive stimulation (ECS), the analogous treatment for rodents, in animal models of depression. In mice treated with corticosterone (a stress model of depression, see Table 1), ECS significantly increased the number of newborn neurons, and more importantly, neurogenesis was required for the antidepressant effect of ECS, since mice lacking neurogenesis did not respond to the therapy [128]. Similar results were obtained in MAP6 knock-out (KO) mice, which share behavioral and neurobiological features of depression, including reduced neurogenesis and altered excitatory and monoaminergic transmission [129]. Interestingly, ECS in these mice not only improved neurogenesis and behavior, but also induced the expression of BDNF. Hence, different classes of antidepressants likely share the same cellular mechanism of action via restoration of adult neurogenesis by BDNF augmentation.

Whereas intact adult hippocampal neurogenesis certainly is required for antidepressant effects, a causative role for neurogenesis in depression is more difficult to be confirmed. Whether a reduction or ablation of adult neurogenesis alone is sufficient to induce depressive-like symptoms is still a controversy, due to contradicting results of diverse studies, which have been extensively discussed elsewhere [40,110,130,131], and will be taken up in Section 5. Nevertheless, since the increase of adult neurogenesis is sufficient to reduce anxiety- and depression-like behaviors [41,42], a positive role of adult neurogenesis in stress-related resilient behavior seems very likely.

### 2.2. MDD and Dysregulated Immune System 

Together with HPA axis overactivation and monoamine dysfunction, dysregulated immune response has been implicated in the pathogenesis of MDD [132,133]. An unbalance between the adaptive and the innate immune response has emerged as a typical immunological signature of MDD [134]. While the number of activated monocytes is increased, T lymphocytes are reduced [135,136]. Consistent with the monocyte activation, circulating levels of proinflammatory cytokines, such as tumor necrosis factor-α (TNF-α), interleukin-1β (IL-1β), and interleukin-6 (IL-6), have been shown to be increased in patients with MDD [137] and with PTSD [138], and in animal models of stress [139,140]. For this reason, cytokine serum levels have been proposed as reliable biomarkers for both MDD and PTSD [138,141]. Noteworthy, both preclinical and clinical studies point to IL-6 as a reliable predictive marker of MDD susceptibility levels. Indeed, higher levels of IL-6 in childhood, likely because of adverse events [142], have been associated with increased risk of depression in adulthood [143] and shown to predict stress resilience in animal models of chronic stress [139]. The pathogenic role of immune dysfunction in MDD is further supported by the results of a large meta-analysis showing that a history of infections or autoimmune diseases is a risk factor for MDD [144].

Hence, it appears that a proinflammatory milieu might be a predisposing factor for later development of MDD and MDD-related suppression of neurogenesis. Proinflammatory cytokines might affect neurogenesis by binding their receptors expressed on both NPCs and neurons, thereby directly regulating NPC fate or by modulating the synaptic inputs onto NPCs, respectively [145]. Indirect mechanisms might also be existing and might arise by the complex relationship between the immune system and HPA axis and the 5-HT biosynthetic pathways, which, as already mentioned, directly modulate neurogenesis. Indeed, consistent with the MDD neuroendocrine and immunological picture, proinflammatory cytokines stimulate GCs and are regulated by GCs [146]. Moreover, proinflammatory cytokines can reduce tryptophan availability in the gut, thus impairing gut microbiota-mediated biosynthesis of 5-HT precursor [147].

## 3. Modulation of Neurogenesis

### 3.1. Positive Modulation of Adult Neurogenesis by Potential Resilience Factors

A variety of factors or conditions upregulating adult hippocampal neurogenesis rate have also been described independently of neurogenesis to be “resilience factors” or to act in an antidepressant manner. This means that mechanistically they could modulate adult neurogenesis to promote stress resilience. In the following, we will summarize what is known about some prominent regulatory factors, such as BDNF and endocannabinoids (eCBs), or conditions, such as exercise and enriched environment in the context of stress resilience by regulating adult hippocampal neurogenesis. In addition, negative modulation of neurogenesis by stress and its disease-promoting role will be delineated (Figure 1).

#### 3.1.1. Environmental Enrichment (EE) and Physical Exercise (PE)

EE and PE are convincingly associated with a broad spectrum of beneficial effects on the hippocampus, including boosting neurogenesis [148]. In animal studies, EE refers to an experimental setting in which rodents are kept in a larger group and in the presence of multiple objects (toys, nesting material, running wheel), to provide animals with social, physical, and cognitive stimulation [149]. PE usually refers to running, mainly performed on a running wheel, to mimic an aerobic activity. Both experimental paradigms are intended to simulate enhanced cognitive and physical stimuli in humans. Despite the lack of direct evidence of improved neurogenesis in humans, an increase of hippocampal volume and cerebral blood flow in this region in people engaged in exercise are reasonably considered suggestive of potentiated neurogenesis in the DG [150]. Pioneering studies published in the late ninetieth of the last century showed that running and EE increase the number of proliferating neurons in the DG [23,151,152], paving the way for flourishing literature on this topic, as nicely reviewed elsewhere [148,149,150,153]. Both paradigms have been shown to influence several aspects of neurogenesis, such as proliferation [154], maturation and morphology [155], and functional integration of newborn neurons [156], which contribute to increased synaptic plasticity in the DG area [151] and improved spatial memory [157]. Acute bout (few days) of running were shown to induce a fast increase of the number of proliferating neurons with prosurvival effects of the progeny [154]. Interestingly, exercise was proven to be the neurogenic component of EE [158,159,160] and to improve neurogenesis even in old animals, counteracting the age- and pathological-dependent neurogenesis reduction [150,154,161]. 

Beyond the peripheral muscle- and endocrine-derived factors, central nervous system (CNS) intrinsic mechanisms have been claimed to play a role in the exercise-mediated proneurogenic effects. Among these, experience-driven increased glutamatergic activity, and upregulation of BDNF levels and signaling are the most accountable [162].

#### 3.1.2. Endocannabinoids (eCBs)

Endocannabinoids (eCBs) are signaling molecules synthesized from membrane lipid components and are derivatives of arachidonic acid, forming the two major eCBs 2-arachidonoyl glycerol (2-AG), and arachidonoyl ethanol amide (also called anandamide, AEA). The high lipophilicity prevents storage in vesicles, and therefore, the intensity of eCB signaling is driven by the activities of the eCB synthesizing and degrading enzymes. eCBs can act in an autocrine and paracrine manner, and are ligands for different receptors, whereby the major receptors are the cannabinoid type 1 receptor (CB1) and type 2 receptor (CB2) [163]. Yet, AEA can also activate TRPV1 (transient receptor potential cation channel subfamily V member 1), while 2-AG also stimulates the GABA_A_ receptor [164,165]. The eCB signaling is involved in many physiological and pathophysiological processes both in the CNS and in peripheral organs [164]. In the context of adult neurogenesis, the research has focused on CB1 and CB2. While these receptors and eCB synthesizing and degrading enzymatic machinery have been reported to be present in NPCs in the SVZ of the adult hippocampus [25,166], the intensity and exact mode of eCB signaling in NPC or onto NPC are difficult to be determined. As these eCB components are additionally expressed in cells surrounding the neurogenic niches in the SGZ, the functionality of eCB signaling regarding the regulation of adult neurogenesis is complex and may act in a paracrine and/or autocrine manner onto neural stem cells and NPCs. 

It has been reported that, in general, eCB signaling, as well as phytocannabinoids regulate adult neurogenesis positively, mostly via CB1 and CB2 [166,167], possibly through multiple mechanisms, including proliferation, antiapoptotic defense, antioxidant defense, immunoregulation, and autophagy/mitophagy [166]. Most of the investigations have addressed these functions under physiological conditions, and only a few investigations addressed stimulated conditions, with positive (e.g., exercise) or negative (e.g., stress) annotation. As discussed above, a link between antidepressant intervention and adult neurogenesis has frequently been reported. In fact, in a mouse model of depressive-like behavior by using CUS, the inhibition of the 2-AG degrading enzyme monoacyl glycerol lipase (MAGL) by chronic application of JZL184 prevented the CUS-induced increase of feeding latency in the novelty-induced suppression of feeding, and immobility time in the forced swim test [168]. The positive behavioral outcome went along with the prevention of CUS-induced impaired adult neurogenesis in the SGZ, and a form of LTP in the DG known to be neurogenesis-dependent. These effects were associated with the normalization of CUS-induced decrease of mTOR (mammalian target of rapamycin) [169]. The mTOR signaling pathway was shown to be compromised in MDD subjects [170], whereas mTOR activation acts in an antidepressant manner [171]. Along with this, activation of mTOR signaling is known to play pivotal roles in adult neural stem cell regulation by particularly upregulating proliferation of the transient amplifying stem cell pool [172], but also by impacting NPC differentiation (for review, see the work by the authors of [173]). A recent other investigation addressed the influence of the microbiome on the eCB system and adult neurogenesis [174]. In an elegant set of experiments using unpredictable chronic mild stress (UCMS) as a mouse model of depression, and fecal microbiota transfer from these mice to non-UCMS mice, the authors rescued the microbiota-transmitted depressive-like behavior by pharmacological inhibition of MAGL with JZL184, concomitantly with the restoration of adult neurogenesis. Furthermore, it was also shown that complementation of UCMS microbiota with *Lactobacillaceae* alleviated depressive-like symptoms and restored neurogenesis levels in recipients of UCMS microbiota.

As outlined above, exercise is an efficient intervention for increasing adult neurogenesis. Pharmacological blockade of the CB1 alleviated the exercise-induced increase in proliferation in the SGZ [175]. In another study, though, using CB1 deficient mice, such a CB1 dependency on neurogenesis was not observed upon a 6-week running period, but the CB1 deficient mice showed reduced motivation to run [176]. The reasons for these divergent observations have not been clarified. 

In summary, the current data on the involvement of the eCB system in stress coping and neurogenesis suggest that the enhancement of eCB signaling, in particular 2-AG, is beneficial for alleviating stress-induced depressive-like behavior, and concomitantly, to the stress-induced blunting of adult neurogenesis. The underlying mechanisms of the stimulatory effects on neurogenesis have still to be further investigated.

#### 3.1.3. Brain-Derived Neurotrophic Factor (BDNF)

The neurotrophin BDNF regulates survival, proliferation, differentiation, and migration of neural stem and progenitor cells in vitro and in vivo during neural development of the embryo, as well as in adult neurogenesis [177,178,179,180]. In mature neurons, BDNF is also well known for its function in synaptic plasticity and LTP formation, thereby controlling cognition, learning, and memory, but also mood [43,181,182,183]. BDNF is secreted at the pre- and postsynaptic side either as proprotein or mature BDNF in an activity-dependent manner or by the constitutive pathway of exocytosis [184,185,186]. BDNF exerts its functions through binding to its two receptors, the high affinity tropomyosin receptor kinase B (TrkB) and the low-affinity p75 pan neurotrophin receptor (p75NTR). Besides being expressed on the vast majority of neurons, the occurrence of both receptor types has been demonstrated in both adult neurogenic niches exhibiting dynamic expression during distinct stages of adult neurogenesis [187,188]. BDNF signaling through the TrkB receptor acts mainly via the PI3K/Akt pathway to positively regulate cellular survival and structural plasticity, whereas the MAP kinase pathway in concert with PLCγ is the main player in regulating cellular proliferation and differentiation. Binding to p75NTR was demonstrated to have opposing functions, e.g., the reduction of dendritic arborization, apoptosis, and long-term depression, also reflecting the enhanced binding of pro-BDNF, for which opposing physiological roles have been demonstrated [189,190,191,192]. 

##### Role of BDNF in MDD

It has been widely shown that serum BDNF availability correlates with mood changes and reflects the pathophysiological state in mood disorders, as well as with structural changes in specific brain regions, such as the hippocampus and cortical areas [193,194,195,196,197]. Moreover, BDNF serum levels seem to reflect BDNF brain levels [198]. Altogether this implicates BDNF as a potential biomarker for MDD, but also for other mood disorders [199]. Indeed, recently, also DNA-methylation profiles of the BDNF promoter were suggested as MDD biomarker, because depressed and healthy individuals could be clearly classified into two groups by this epigenetic modification [200]. The BDNF hypothesis of depression is justified because opposing actions of stress and antidepressant treatment are observed on existing BDNF levels in serum and limbic brain regions, such as the hippocampus [182]. Stress significantly suppresses mRNA and protein BDNF levels in the hippocampus, particularly in the DG and CA3 hippocampal subfields, and thereby impairs downstream targets of signaling pathways implicated in neuroplasticity [201,202]. Two important meta-analyses could directly prove decreased serum BDNF levels in depressed, suicidal patients, whereas BDNF was increased after antidepressant treatment in humans [195,196]. The question of how BDNF exerts its antidepressant effect is still not fully understood, since the regulation by BDNF could appear at the level of neuronal excitability, as well as regarding the regulation of adult neurogenesis or both. Furthermore, brain atrophy caused by stress [203] could be potentially counteracted by BDNF, serving as a survival factor for degenerating neurons. However, this last point is unlikely because some antidepressants reported an increase of BDNF that did not reverse stress-induced atrophy [182,203]. 

##### Role of BDNF in Neurogenesis Regulation

The discovery that most classical antidepressants, such as SSRIs, norepinephrine reuptake inhibitors (NERI), or monoamine oxidase inhibitors (MAOs) under chronic administration not only increase BDNF expression and signaling, but are also strong inducers of adult neurogenesis [43,204,205], finally led to the neurogenesis hypothesis of depression, whereby BDNF is a central player (see Section 2.1). In fact, infusion of BDNF into the hippocampus of mice mimics the effects of antidepressants in behavioral tests and on neurogenesis rate [206]. Furthermore, in mice with compromised or selectively ablated BDNF/TrkB signaling, antidepressants failed to induce both neurogenesis and improved behavior in mood tasks [207,208]. On the other hand, the ablation of proliferating neural stem and progenitor cells could demonstrate the requirement of hippocampal neurogenesis for the behavioral effects of antidepressants [116,117]. 

The positive regulation of BDNF on adult neurogenesis is, on one hand, a survival effect. This is because heterozygous BDNF mice or mice heterozygous for TrkB displayed a reduced survival of newborn neurons without changing proliferation rate of neural stem or progenitor cells [208], although specific TrkB deletion on neural progenitors was shown to decrease proliferation [209]. The survival function of BDNF was also demonstrated for the EE paradigm, known to upregulate BDNF and neurogenesis; likewise, the survival of newborn neurons was not augmented in heterozygous BDNF mice [210]. On the other hand, BDNF promotes differentiation and maturation of neural stem and progenitor cells through the involvement of GABAergic transmission from local interneurons in the hilus of the DG [211]. As mentioned above (see Section 3.1.1), besides antidepressants and EE, also exercise, specifically running, induces neurogenesis via increased BDNF availability [212,213,214].

##### BDNF as a Mediator of Stress Resilience

The question of whether individual BDNF expression levels can prevent susceptibility to stress or lead to resilience has been addressed by one study in rats using localized BDNF overexpression or knockdown in the hippocampus weeks before the chronic mild stress (CMS) paradigm. Indeed, Taliaz et al. (2011) reported that individual high BDNF levels consequently lead to a higher degree of stress resilience coupled to increases in neurogenesis. BDNF-mediated stress resilience to learned helplessness (LH) was also demonstrated by individually higher expression in the hippocampus of resilient than in susceptible rats [215]. Interestingly, also acute effects of amino acid metabolites-induced BDNF/TrkB signaling led to stress resilience in a mouse model of CSDS [216]. Vice versa, mice deficient in BDNF or with decreased BDNF/TrkB signaling are more susceptible to acute mild stress, subchronic mild stress and CSDS, by displaying increased plasma corticosterone levels [217,218,219,220]. In the CUS model, however, one study reported that deficits in BDNF did not increase vulnerability to stress, but nonetheless dampened its antidepressant-like effects [221]. 

Genetic association studies in humans predict the occurrence of the Val66Met polymorphism of the *BDNF* gene as a risk factor for MDD [222,223]. The BDNF Val66Met variant alters intracellular trafficking and activity-dependent secretion of BDNF, leading to reduced BDNF function associated with decreased exercise-induced neurogenesis rate in mice [224,225]. 

Altogether, the mentioned publications suggest BDNF as a potent resilience factor via the regulation of adult neurogenesis and consequently by inducing behavioral changes in an antidepressant-like manner. 

### 3.2. Negative Modulation of Adult Neurogenesis by Stress

Long-term exposure to environmental, physical, and psychosocial stress is a recognized risk factor for MDD, also referred to as stress-related disorder [132]. A plethora of stressors contributes to the development of MDD, including traumatic events, such as bereavement, repetitive job hassles, diagnosis of a disabling disease, physical or sexual abuse. The time-window of trauma exposure has a leading role in determining the body’s structural and functional changes in response to stress. In this respect, early life stress (ELS), such as childhood trauma (for example, abuse), lack of maternal care, poor nutritional intake, triggers significant changes in the brain with psychological consequences in adulthood [226]. The hippocampus, which mostly develops postnatally in both humans and rodents [227,228], is highly sensitive to precocious stress. ELS in rodents was shown to impair adult neurogenesis, in correlation with impaired learning and memory functions (reviewed by the authors of [226]) specifically in male rodents [229,230], reviewed by the authors of [231].

From a neuroendocrine point of view, acute stress engages a fast and self-limiting body reaction that implicates the involvement of the stress hormones, cortisol, norepinephrine, and epinephrine, the immune system, and stress-sensitive brain areas, such as the hippocampus. The complex interaction among these factors underlying the so-called “fight or flight response” is a beneficial protective mechanism that prepares the body to react to stressors [232]. A crucial role in the stress system is played by GCs and the HPA axis. Activation of the HPA axis starting from the release of corticotropin-releasing hormone (CRH) from the hypothalamus to stimulate the pituitary release of adrenocorticotropin hormone (ACTH) leads to the final synthesis and release of cortisol in humans and corticosterone in rodents from adrenal glands [146]. GC levels, in turn, block the HPA axis, through negative feedback over the hypothalamus, and as mentioned above, the hippocampus. This area is particularly rich in GC receptor (GR), which, in contrast to the other GC responsive receptor, the mineralocorticoid receptor (MR), has been implicated in the negative feedback to stress [233]. 

Prolonged exposure to stressors and/or the lack of efficient termination of the stress response can lead to maladaptive changes in the whole stress response system, which ultimately give rise to stress-related diseases. The individual susceptibility or resilience to stress depends on several intrinsic factors (genetics) and external (environment, lifestyle) that allow a passive or an active coping behavior [234]. Coping behavior implies cognitive and emotional processing of experiences, which, as mentioned above, fits well to hippocampal- and neurogenesis-dependent functions, such as pattern-separation and behavioral flexibility [17]. The direct mechanisms of GCs on adult neurogenesis are described in detail in Section 4.1. 

As discussed above, several animal models have been developed to study the neuroendocrine and neuroimmune responses, as well as adaptations to chronic stress. Moreover, these models, except for the lipopolysaccharide (LPS) model, which is based on the inoculation of an inflammatory agent, are designed to cover the whole range of different human adverse experiences, ranging from early life trauma to physical and psychosocial stress (see Table 1 and the work by the authors of [57]).

## 4. Mechanisms of Stress Acting on Neurogenesis

To date, it is mechanistically not clarified how stress leads to suppression of adult neurogenesis in the DG. Generally, reduction of newborn neurons could arise from the decreased proliferation of neural stem/progenitor cells or diminished survival, which could occur for each type of stem/precursor cell during the whole course of neuronal differentiation until final maturation. The obvious cell death mechanism is apoptosis, since it has been shown that adult neural stem cells and their progeny can exhibit the machinery of the apoptotic cell death program. Furthermore, it is well known that approximately 50% of neural stem cells and early NPCs physiologically die by apoptosis and that the extent depends on the availability of certain growth factors, such as BDNF, vascular endothelial growth factor (VEGF), fibroblast growth factor 2 (FGF2), and epidermal growth factor (EGF) [19,235]. Recently, new concepts of cellular degradation in adult neurogenesis appeared by demonstrating the occurrence of autophagy in regulating adult stem cell homeostasis and neuronal morphogenesis [236,237,238,239]. For example, mice knocked-out for Atg5, a key autophagic factor, displayed delayed maturation and reduced survival of adult-born neurons [240]. Autophagy is principally regarded as a physiological cytoprotective mechanism via regulated degrading and recycling of unnecessary or dysfunctional components. However, in disease, autophagy is observed to be not only part of the adaptive cellular response to stress, but also appears to promote cell death. Very recent data showed that specific deletion of Atg5 in adult neural stem cells prevented cellular decline after chronic restraint stress in mice [241]. Interestingly, in this study, no apoptotic signs in degenerating neural stem cells and their progeny were detected after the stress procedure. This is in contrast to other studies which widely detected apoptosis in the SGZ after different protocols of stress exposure [242,243,244]. Therefore, in the future, it remains to be determined to which extent neural stem cells and their progeny are affected by which mode of cellular death under certain stress conditions. In the following chapter, we will discuss the cascades of events from the starting point of prolonged high concentrations of GCs to the endpoint of reduced numbers of newborn neurons, summarized in Table 2 and Figure 2.

### 4.1. The Role of HPA Axis and GCs in Stress-Induced Reduction of Hippocampal Neurogenesis 

#### 4.1.1. The Complex Interplay between GCs and Hippocampal Neurogenesis

The persistence and nature of the stressor can cause the dysregulation of the HPA axis with chronically increased GC levels and neurogenesis modulation [258]. Although a large number of data supports the causal link between elevated GCs and impaired hippocampal neurogenesis, especially in stress-related psychiatric disorders, it should be noted that elevations in GC levels have been described in rodents exposed to the proneurogenic factors, EE, and voluntary exercise [259,260,261,262]. Thus, different stimuli with opposing effects on neurogenesis can converge on the same effector molecule, the GC, likely involving additional and intermingled mechanisms, such as the serotoninergic and glutamatergic system and the neurotrophins, as nicely reviewed by Saaltink and Vreugdenhil [263]. Interestingly, Lehmann and colleagues showed that EE can restore normal behavior and improve neurogenesis in defeated mice by releasing GCs [264]. Indeed, the authors provided convincing evidence in mice that neurogenesis is the crucial mediator of GC-induced depressive-like behavior, since adrenalectomy before CSDS improved behavioral outcome and neurogenesis, whereas neurogenesis ablation prevented the protective effects of adrenalectomy. Moreover, indeed, adrenalectomy followed by EE in previously defeated mice prevented the proresilient and proneurogenic effects of EE [264]. These results suggest that EE can act as a therapeutic/proresilient tool in stress-related disorders through a fine-tuning regulation of the HPA axis.

Multifaceted and bidirectional interactions between GCs and neurogenesis have emerged in healthy and stress conditions, thus making this issue complex and worth to be investigated. The antineurogenic role of GCs is well documented in a variety of animal models. Chronic corticosterone treatment has been shown to impair adult neurogenesis [84,265,266] in a sex- and administration method-dependent manner [267,268] and with a specific effect in the ventral hippocampus [269]. Noteworthy, adrenalectomy has been shown to prevent the age-associated upregulation of the HPA axis and the impaired neurogenesis in adults [270,271]. Interestingly, adrenalectomy delivered at postnatal stages was shown to induce a transient effect on neurogenesis, no more detectable in the adult hippocampus, suggesting that other long-term compensatory mechanisms may take place [272]. In line with this, adult rats with chronically low levels of GCs showed an untouched neurogenesis rate, whereas adrenalectomy boosted adult neurogenesis [273]. These data suggest that the depressant effects of GCs on neurogenesis are temporary and can be easily reversed. In addition, it should be noted that adrenalectomy induces extensive apoptosis in the DG, selectively affecting old granule cells [274], with the induction of irreversible spatial memory deficits [272]. On the other hand, spatial memory has been linked to an active selection and removal of different populations of newly born neurons, likely those not fully integrated into neuronal circuits [275]. Thus, GC levels might be involved in modulating neuronal circuitry in the DG and neurogenesis, thereby, regulating learning.

#### 4.1.2. Evidence for GR Involvement in Stress-Induced Hippocampal Neurogenesis Reduction

Several studies have dissected the contribution of the GC system to the behavioral and neurogenic sequelae of stress, focusing on the role of GRs. Indeed, GCs control the neurogenic niche, mainly through the GRs [276], since MRs are not expressed by NPCs [277]. Interestingly, Fitzsimons and colleagues showed that GR knock-down in the DG neurogenic niche increased the number of doublecortin (DCX)+ neuroblasts, accelerated their terminal differentiation, and increased basal excitability, in line with the idea that GCs can impair neurogenesis, by altering the excitation-inhibition balance [276]. More recently, it has been shown that NPCs isolated from the rat ventral hippocampus are more sensitive to the antineurogenic effects of chronic GC treatment compared to the dorsal and the intermediate hippocampus, thus explaining the enhanced susceptibility of NPCs of this hippocampal region to the stress [86]. In the context of stress, heterozygosity for GR was shown to make mice prone to develop depressive-like behavior and impaired neurogenesis after stress [278,279]. Hence, increased sensitivity of NPCs to GCs may influence stress response. This issue has been investigated in ELS models. Maternal separation was shown to accelerate the age-dependent increase of corticosterone, neurogenic suppression, and depressive-like behavior in the adult offspring [280]. In contrast to these findings, repeated maternal separation was reported to cause an early suppression of neurogenesis, detectable in adolescent mice without significant changes in corticosterone levels [93]. Moreover, in the adult rats previously exposed to the repeated maternal separation, increased levels of corticosterone and remarkable depressive-like behavior were observed, suggesting that ELS can predispose to develop emotional alterations, by preconditioning neurogenesis ontogeny [93]. These data highlight the complexity of the interplay between adult neurogenesis and the GC system and support the bidirectionality of this relationship. In fact, hippocampal neurogenesis itself can control the HPA axis. Neurogenesis blockade was shown to either increase GC levels under mild and acute stress [38,281] or decrease GC levels in the restraint test [131]. Vice versa, transgenic mice showing improved neurogenesis were protected against UCMS-induced neurogenic suppression and showed decreased negative feedback of the HPA axis compared to control mice [42]. 

#### 4.1.3. Targeting the HPA Axis as Proresilience and Proneurogenic Factor

Building on the recognized leading role in stress response, the HPA system has been investigated as a potential target to promote neurogenesis and resilience to stress. Repeated administration of an antagonist of the CRH receptor reversed the neurogenesis impairment caused by CMS, similarly to the antidepressant fluoxetine [282]. Treatment with GR antagonists has been shown to rescue the neurogenesis suppression induced by exposure to GCs or chronic stress [283,284,285]. Through a peptide array analysis, a selective modulator for GR was identified and shown to have differential brain effects, behaving as a partial agonist for the suppression of CRH gene expression and contrasting the GR-mediated reduction of hippocampal neurogenesis after chronic corticosterone exposure [286]. Compelling data have been provided in studies using animal and human cells in vitro. Exposure of human hippocampal fetal NPCs to corticosterone induced lasting changes in DNA methylation, which resulted in the enhanced transcriptional response of specific DNA sequences upon GC re-exposure [287]. The researchers used these in vitro results to compute a GC-responsive poly-epigenetic score of the differentially methylated sites. Noteworthy, analysis of newborn’s cord blood DNA showed significant associations between this score and maternal depression and anxiety [287]. Likewise, in another study, the serum- and GC-inducible kinase 1 (SGK1), a GC target gene with relevant implications for MDD pathogenesis [288], was identified as a crucial molecular mediator of GC-dependent neurogenesis impairment in in vitro cultured human NPCs [245]. Importantly, SGK1 mRNA levels were increased in the peripheral blood of drug-free depressed patients and in the hippocampus of rats that underwent either prenatal stress or UCMS [245]. 

GC levels have been shown to underlie different MDD phenotypes, with high-GC subjects expressing a melancholic depression and low-GC subjects atypical MDD. Based on this, a mouse model characterized by low-GCs and high-GCs, respectively, has been established to replicate distinct behavioral and neurogenic phenotypes of MDD in mice [289]. Interestingly, fluoxetine effects were shown to depend on the GC endophenotype in this model [290], with a reversal of the behavioral phenotype (increased active coping) and the increase of neuroblast number in high-GC mice and an exacerbation of the behavioral despair (increased passive coping) and suppression of DG cell proliferation in low-GC mice. These results highlight that specific endophenotypes of the GC system can influence the efficacy of antidepressive treatments, likely accounting for individual responses to fluoxetine.

Despite the complexity of the neurogenesis-GC interplay described here, most of the data in the literature point to GCs as crucial players in determining neurogenesis response to stress, acting as an upstream regulator.

### 4.2. GC-Induced BDNF Decrease, Impaired BDNF/TrkB Signaling

Under physiological conditions, the source of BDNF in the DG derives from mature DG granule neurons to stimulate neurite outgrowth and ramification of the dendritic tree [211,291]. In addition, BDNF is secreted from the entorhinal cortex (EC) to the DG by axons of the perforant path, which constitutes the main structural and functional input to the hippocampal formation [292]. The mechanism of how chronic stress lowers BDNF availability in the DG, and hence, decreases adult neurogenesis is still not completely understood. There are multiple levels of GC regulation of BDNF, whereby the three main levels are: (1) transcriptional regulation, (2) signaling, and (3) transport. 

#### 4.2.1. GC-Mediated Transcriptional Repression of BDNF

Transcriptional downregulation of the *BDNF* gene is the most obvious regulation by GCs, since GCs act via steroid hormone receptors, which are ligand-activated transcription factors that directly bind to GC response elements (GRE) in the genome. The direct repression of the BDNF gene by GCs was long discussed and demonstrated indirectly; however, the clear molecular mechanism and the occurrence of putative GREs were longtime missing. In 2017, Chen et al. [293] demonstrated for the first time direct GR binding to BDNF regulatory sequences in vitro when using cultured neuronal cells. Nevertheless, it cannot be excluded that other GRE regulated factors contribute to the modulation of BDNF transcription, for example, the activator protein-1 (AP-1) complex or cAMP-response element binding protein (CREB), an important positive regulator of BDNF expression (reviewed by the authors of [294]). 

#### 4.2.2. GC-Mediated Compromised BDNF/TrkB Signaling

Chronic GCs could furthermore modulate the expression of BDNF receptors, which would lead in consequence to decreased BDNF-signaling in NPCs and newborn neurons. Whereas, GC-reduced expression of TrkB receptor mRNA seems unlikely [295], it is nonetheless conceivable that changes in the molecular ratio of truncated and catalytic TrkB or expression changes in the mostly understudied p75NTR are present [296]. Compromised BDNF signaling could additionally arise from GC effects on proteolytic cleavage of BDNF by changing levels of intracellular and extracellular proteases [297,298], and thus, diminish the availability of mature BDNF, whereby increasing pro-BDNF. The imbalance of the pro- and mature forms of BDNF has been observed in depressed patients and in rodent models of depression [299,300,301]. Furthermore, antidepressants were shown to correct this imbalance in the brain of chronically stressed mice [302]. GCs could further directly interrupt BDNF signaling, by inhibiting the prosurvival (PI3K/Akt; PLCγ) and proliferative (MAP kinase) pathways. Indeed, in NIH-3T3 fibroblasts, GCs upregulate the MAP kinase inhibitor protein MAP kinase phosphatase 1 (MKP1), a potent terminator of MAP kinase signaling [303]. In addition, in cultured cortical neurons, GCs have been shown to hinder the direct interaction of TrkB and GR receptors, which is important for the induction of the PLCγ pathway [304]. It is obvious that these data need to be proven in vivo by analyzing whether chronic stress impairs BDNF signaling directly in hippocampal NPCs and their progeny. 

#### 4.2.3. GC-Induced Decrease of Axonal Transport of BDNF

Interestingly, it was shown that electrical stimulation of the entorhinal cortico-hippocampal circuit alleviated depressive-like symptoms in mice after chronic stress exposure by augmenting adult neurogenesis [305,306,307]. In a recent publication of Agasse and colleagues, the authors demonstrated that corticosterone slows cortical transport of BDNF vesicles via cyclin-dependent kinase 5 (Cdk5)-dependent hyperphosphorylation of huntingtin (htt), which leads to suppression of adult neurogenesis [246]. Impaired transport of BDNF vesicles along microtubules has already been attributed to the misfunctioning (mutated) htt protein in the neurodegenerative Huntington’s disease (HD) [308,309]. Interestingly, patients with HD often suffer from psychological impairments resembling MDD long before locomotor deficits manifest, and at least rodent mouse models of HD display reduced hippocampal adult neurogenesis [310,311]. Therefore, a common mechanism in reducing neurogenesis by impaired htt-mediated BDNF-transport might explain mood disturbances in both diseases HD and MDD. The htt-phosphorylating kinase Cdk5 has already been implicated in regulating embryonic and adult neurogenesis, in which it has, on the one hand, a maturating and survival-promoting role, but when dysregulated, it can induce cellular death [312,313]. Interestingly, its activity is increased in various brain areas of the limbic system in response to stressors, and therefore, it has been linked with neuropsychiatric, but also neurodegenerative diseases [314]. In fact, infusions of a Cdk5 inhibitor into the hippocampal DG, but not CA1 or CA3, increased sucrose preference and prevented locomotor impairment in response to CMS, supporting antidepressant activity [315]. Interestingly, Cdk5 is also an important kinase participating in hyperphosphorylation of the microtubule-associated protein tau in Alzheimer’s disease, where impaired neurogenesis and depressive-like symptoms are similarly found as in the case of HD [316,317]. In fact, chronic stress exposure can also lead to tau hyperphosphorylation [318], which on the one hand, is known to affect axonal transport, as well as neuronal plasticity by deficits in the cytoskeletal architecture [319]. Indeed, tau was described to have key functional roles in the morphogenesis of newborn hippocampal neurons [320]. Recent publications demonstrated a tau-dependent suppression of neurogenesis in the stressed hippocampus [82,321]. Stressed tau KO mice did not exhibit a reduction in the DG of proliferating cells, neuroblasts, and newborn neurons, which the authors attributed to retained PI3K/mTOR/GSK3β/β—catenin signaling in mediating survival and proliferation in neural stem and progenitor cells via putatively induced BDNF expression [322]. Furthermore, addressing tau’s role on microtubules, it is conceivable that stress-induced hyperphosphorylated tau decreases axonal transport of BDNF vesicles from the EC to the DG, as it has been shown for stress-induced hyperphosphorylated htt [246].

### 4.3. Serotonin (5-HT)/Signaling Reduction

#### 4.3.1. Serotonergic Regulation of Adult Neurogenesis

Adult neural stem cells in the SGZ strongly depend on serotonergic innervation by projections from the median and dorsal raphe nuclei (RN). In fact, lesion of the dorsal raphe projections to the DG leads to a decrease in adult neurogenesis [323]. Vice versa, KO mice for the 5HT-transporter 5HTT (alternatively named as Sert), which removes 5-HT from the synaptic cleft, increases the production of new neurons. In line with this, antidepressants blocking the monoamine degrading enzyme MAO or 5HTT upregulate neurogenesis [324]. For example, chronic treatment with the SSRI fluoxetine increases survival and maturation of NPCs and newborn postmitotic neurons, thereby inducing an augmentation of net neurogenesis [325]. A further interesting aspect is the requirement of 5-HT for the exercise-induced upregulation of adult neurogenesis, which occurs through a proproliferative effect on neural stem cells [326].

Serotonergic regulation is involved at all levels of adult neurogenesis, proliferation, differentiation, maturation, and survival, and executed by the concerted action of a bulk of different 5-HT receptors in the DG appearing on distinct cell types. Whereas, mature granule cells express 5-HT_1A_, 5-HT2_B,C_, and 5-HT_4_ receptors, radial-glia-like (RGL) type-1, and transient amplifying type-2 neural stem cells express exclusively 5-HT_1A_ receptors. In contrast, hilar interneurons express besides 5-HT_1A_, also 5-HT_2A_, and 5-HT_3_ receptors [112,247,248,325].

#### 4.3.2. 5-HT Receptors in MDD and Their Function in Adult Neurogenesis

Surprisingly, the 5-HT deficiency theory in depressed patients still remains controversial after years of extensive research. The major problem is that 5-HT levels so far can be only measured in the postmortem human brain, and tissues of animal studies are not always reliable indicators of extracellular levels [110,327,328,329]. Nevertheless, increasing 5-HT levels by antidepressants strongly implicate involvement of the serotonergic system in MDD, which does not necessarily need to be causative for the disease. Moreover, one could argue that impairing serotonergic signaling, by e.g., chronic stress, could also appear in dysregulated expression or de-/sensitization of 5-HT receptors. Indeed, here, data obtained by specific receptor gene deletion and/or pharmacological intervention in animal rodent models of depression are more conclusive. Accumulating evidence indicates a role in MDD for at least 5 of the 14 5-HT receptor subtypes: 5-HT_1A_, 5-HT_1B_, 5-HT_4_, 5-HT_6_, and 5-HT_7_ [330,331]. A particular focus lies on 5-HT_1A_ receptors, for which human genetic and imaging studies revealed differences in their expression levels and regulation during the course of MDD and antidepressant medication [332,333]. Furthermore, the occurrence of the C(-1019)G polymorphism in the promoter region of the 5-HT_1A_ receptor gene (*HTR1A*) is associated with MDD and response to antidepressant treatment [334].

5-HT_1A_ receptors (5-HT_1A_Rs) exist in two distinct receptor populations, either as somatodendritic autoreceptors on 5-HT producing neurons located in the RN or as postsynaptic heteroreceptors. The 5-HT_1A_R heteroreceptors mediate local neuromodulatory effects in multiple brain areas innervated by serotonergic projections, including the DG [335,336,337,338]. Considering the role of the ventral DG in emotional regulation, it is very interesting that particularly the expression of 5-HT_1A_R increases along the dorsoventral axis with the highest expression levels at the ventral pole [339]. In addition, a decrease of 5-HT_1A_R expression in the DG by corticosterone has been demonstrated in rodents [340,341]. Depletion of 5-HT and simultaneous activation of 5-HT_1A_R by the specific agonist 8-OH-DAPT resulted in increased proliferation of adult progenitor cells [342], whereby also stimulation of 5-HT_1A_R alone was sufficient to increase the neurogenesis rate [116,343]. Consistent with this, specific 5-HT_1A_R blockade or mice germline deficient for 5-HT_1A_R display reductions in neurogenesis and do not respond to SSRIs [116,344]. Importantly, the proliferative effect of 5-HT_1A_R activation was also demonstrated in vitro in neurosphere culture, which demonstrates that direct and indirect effects on neuralstem/progenitor cells can occur in parallel, an issue which is still a matter of research [325]. 

#### 4.3.3. Neurogenic Growth Factor Support by 5-HTRs

Interestingly, specific deletion of 5-HT_1A_R on mature, but not on young adult-born granule cells ablated the neurogenic and behavioral response to SSRIs. In line with this, re-expression of 5-HT_1A_R exclusively on mature DG granule cells on a 5-HT_1A_R deficient background was sufficient to mediate the neurogenic and antidepressant response of SSRIs [247]. Furthermore, the SSRI-induced increase of BDNF and VEGF was only attenuated when specifically knocking down 5-HT_1A_R in mature granule cells. These data suggest that particularly mature granule neurons mediate the antidepressant response by 5-HT_1A_R stimulation. Nevertheless, the involvement of astrocytic 5-HT_1A_R with subsequent release of neurotrophic factors cannot be excluded [345,346]. The general question of how growth factor support is mediated by 5-HT_1A_R signaling is so far not elucidated. 5-HT_1A_R is an inhibitory G-protein-coupled receptor, which, once activated, leads to cAMP decrease. However, the expression of both BDNF and VEGF is dependent on CREB-binding to the cAMP-response elements (*CRE*) in their promoter region. Nevertheless, since 5-HT_1A_R was demonstrated to activate also other signaling cascades involving, for example, ERK and Akt, which can lead to CREB activation, it is possible that induction of *CRE*-mediated transcription could occur [347]. In this context, the involvement of 5-HT_4_R, highly expressed in the mouse DG, is more evident [248,339,348]. 5-HT_4_R is a G_s_-coupled receptor, leading to increased cAMP levels after 5-HT binding and leads directly to enhanced growth factor expression and secretion via cAMP/PKA-activated CREB [349,350]. Pharmacological studies demonstrated that 5-HT_4_R activation leads to enhanced proliferation of neural stem/progenitor cells and maturation of newborn neurons [349,350]. In line with this, genetic deletion or chronic inhibition decreases adult neurogenesis, and it partially blocks the effects of the antidepressant fluoxetine [248,351]. In the DG, 5-HT_4_R expression seems to be limited to mature granule cells, as DCX+ neuronal progenitors, as well as calretinin+ immature granule cells, did not reveal beta-galactosidase reactivity in a 5-HT_4_R reporter mouse line. However, 5-HT_4_R expression in adult neural stem cells was not directly addressed in this study by Imoto et al. (2015); only a lack of signal in the SGZ by in situ hybridization was reported [248]. So far, no study exists describing 5-HT_4_R DG cell-type specific deletion to prove whether, indeed also here, particularly mature granule cells are necessary for the neurogenesis-driving action of 5-HT_4_R. 

Interestingly, it was observed that antidepressant treatment, but also ECS, leads to a phenomenon called dematuration of previously mature granule cells, which obtain an immature granule cell phenotype with the characteristic electrophysiological properties [248,352]. In addition, these dematuration processes were abolished in 5-HT_4_R KO mice [353]. Thus, the antidepressant response by 5-HT_4_R could act either through increased neurogenesis or dematuration, which is an interesting alternative to the common neurogenesis hypothesis of SSRI action [354].

### 4.4. Excitation/Inhibition Imbalance 

Accumulating evidence exists that excessive glutamatergic neurotransmission contributes to the etiology of depression [355,356,357]. Antidepressant treatment with classical antidepressants targeting the monoaminergic system was shown to decrease glutamate levels in depressed individuals and normalize AMPA receptor signaling, which accounts for decreased inhibition of glutamate release by 5-HT during depressive states [358,359]. In addition, more direct therapeutic interventions by antagonism of NMDAR, e.g., by ketamine or memantine, are supposed to be an effective pharmaceutical mode of antidepressant action with behavioral and neurogenic improvements [122,360,361]. 

Interestingly, the presynaptic metabotropic glutamate receptor mGluR2, an inhibitor of synaptic glutamate release, was identified as a marker of stress susceptibility [362]. It was demonstrated that the individual stress response correlates with MR-regulated low expression levels of mGluR2, which decreases resilience to stress [363] and is associated with dendritic loss and reduced DG neurogenesis [364].

#### 4.4.1. Regulation of Adult Neurogenesis by Excessive Glutamate

At the cellular level, glutamate has a biphasic role depending on its concentration, and hence, its impact on neurotransmission. Although low glutamate excitation of NMDAR generally favors cellular survival via upregulation of BDNF [365], high glutamate-induced NMDAR activation, e.g., by chronic stress, is neurotoxic through calcium ER stress and prevents BDNF expression [366,367]. Likewise, regulating adult neurogenesis by glutamate seems biphasic depending on glutamate concentrations. On the one hand, NMDAR activation on proliferating progenitors by low glutamate tone leads to neuronal differentiation in vitro [368] and decreased NMDAR expression in mice to impaired neurogenesis [369]. On the other hand, excessive NMDAR activation, e.g., after prolonged GC exposure, inhibits neurogenesis [270,370,371], presumably due to excitotoxicity-induced cell death. It might be hypothesized that a biphasic and time-dependent effect of glutamate on neurogenesis occurs in MDD, similarly to epilepsy [372]. 

#### 4.4.2. GABAergic Dysregulation in Neurogenesis and MDD

Elevated glutamate levels and excessive glutamatergic neurotransmission after prolonged GC exposure definitively point towards an insufficient GABAergic inhibition of glutamatergic neurons. In fact, GCs modify GABAergic transmission via modulation of GABA release and uptake [373,374], binding to GABA_A_ receptors [375], and furthermore dysregulate expression of GABA_A_ receptor subunits [376,377]. MDD patients display reduced GABA levels in the cerebrospinal fluid [378], plasma [379,380], and in the brain [381,382], whereas SSRI or ketamine treatment normalize GABA deficits of patients [383,384,385,386]. Furthermore, postmortem brain analysis of MDD patients revealed a reduction of calbindin+ GABAergic interneurons in the prefrontal and occipital cortex [387,388]. In the DG, a decrease in calretinin+ and parvalbumin (PV)+ interneurons was observed in rats that had undergone CMS [389], and interestingly, in chronically stressed shrews, the deficit in DG PV+ interneurons was prevented by fluoxetine [390]. PV+ interneurons appear to be a particularly vulnerable population in chronic stress [391], and furthermore, they are important regulators in the neurogenic niche of the adult DG [392]. PV+ interneurons regulate the quiescence of type-1 RGL stem cells in an activity-dependent manner. Heightened activity of PV+ interneurons inhibits quiescent stem cell activation and promotes survival of proliferating NPCs. With low activation, for example, seen in a social isolation (SI) paradigm of chronic stress, an expansion of the type-1 stem cell pool is observed at the expense of neuronal production, since suppressed survival of dividing NPCs occurs in parallel [249]. Another study reported that conditional heterozygous deletion of γ2 subunit-containing GABA_A_ receptors on postmitotic immature neurons in the adult DG led to decreased adult neurogenesis by reduced differentiation, maturation, and/or cellular survival and was associated with increased behavioral responses to stress [250]. 

Altogether, it is conceivable that the dysfunction of reduced GABA signaling as seen during chronic stress leads to disturbances of adult neurogenesis by directly affecting adult neural stem cells and their progeny. Additionally, reduced GABAergic inhibition of mature granule cells leading to heightened glutamatergic signaling can indirectly reduce adult neurogenesis through elevated glutamate signaling. Furthermore, DG glutamatergic granule cells and GABAergic interneurons in the DG are both innervated and regulated by serotonergic presynapses from the raphe nuclei, which would account for a superior level of neurotransmitter imbalance during stress (see Section 4.3).

### 4.5. The Role of Proinflammatory Cytokines in Stress-Induced Hippocampal Neurogenesis Modulation

#### 4.5.1. Proinflammatory Cytokine Control of Hippocampal Neurogenesis

Proinflammatory cytokines have been widely demonstrated to have a causative role in depressive behavior in both human and animal settings [393]. Noteworthy, in the LPS model, inflammation has been shown to directly affect hippocampal neurogenesis [104,105,106,107]. Moreover, prenatal or postnatal LPS administration has been shown to reduce neurogenesis and induce depressive-like behavior in adulthood [394,395,396,397], supporting the notion that inflammation might predispose to neurogenic and behavioral deficits associated with depression. Potentiation of IL-1β [398,399,400,401], TNF-α [402,403], and IL-6 [404,405] signaling using transgenic mice or in vivo administration of single cytokine, has been generally associated with poor neurogenesis in rodents. Interestingly, a huge bulk of data in cultured NPCs suggest that cytokines can exert both pro- and antineurogenic effects in a dose-dependent manner [406,407,408], supporting the notion that physiological release of cytokines controls brain functioning at several levels [409]. 

#### 4.5.2. Neurogenic Inhibition by Stress-Related Proinflammatory Cytokines 

Acute and chronic stress induces an inflammatory response, followed by raises in proinflammatory cytokine levels (for an exhaustive review, see the work by the authors of [258]). Chronic stress in adult mice has been associated with a peripheral increase of IL-6, TNF-α, and IL-1β and impaired neurogenesis [140]. Prenatal stress delivered in pregnant rats through sleep deprivation was shown to reduce both BrdU+ and DCX+ neurons in the hippocampus of young offspring in association with increased IL-6, TNF-α, and IL-1β expression and microgliosis, but data on long-term effects have not been provided [410]. In this context, IL-1β has been proposed to play a role in sex-dependent differences in the rate of neurogenesis observed in rats, and to contribute to impaired neurogenesis in adult rats of both sexes born from stressed pregnant rats [97]. Natural compounds with anti-inflammatory action have been proven effective in reversing stress effects on both behavioral and neurogenesis-related outcomes in rodent models of stress, with significant reductions of IL-6, TNF-α, and IL-1β [75,411,412,413].

In the wake of the increasingly recognized role of cytokines in stress response, some studies have specifically targeted single cytokine in rodent models of stress, mainly focusing on IL-1β, since this cytokine is significantly upregulated in the hippocampus of stressed mice [258]. Despite the recognized role of peripheral IL-6 levels in predicting stress susceptibility in rodents [139,414] and MDD risk in humans [143], to our knowledge, the involvement of IL-6 in chronic stress-mediated neurogenesis modulation has not been explored so far. It should be noted that IL-6 has been implicated in LPS-induced depression of neurogenesis in the LPS model [397], which, however, is a pure inflammatory model and does neither fully replicate the behavioral sequelae of stress and depression nor discriminate between susceptible and resilient subjects. Regarding TNF-α, in a rat corticosterone-induced depression model, peripheral administration of the TNF-α inhibitor etanercept prevented the loss of newborn neurons, promoted the complexity of the dendritic branching of the new neurons, and recovered hippocampal-dependent memory deficits [415]. 

#### 4.5.3. The Crucial Role of IL-1β in the Stress-Induced Neurogenic Response

Several studies have shown that IL-1β is upregulated in the hippocampus of several stress models, providing that this cytokine is a good target in relieving stress-induced behavioral and neurogenic depression [258]. Intra-cerebroventricular infusion of IL-1β mimicked the effects of acute stress (foot-shock or immobilization) on the proliferation of precursor cells in the DG of adult rats. Exposure to CUS blocked both the proliferation of precursor cells and the formation of neuroblasts, while chronic blockade of IL-1β recovered the antineurogenic effects of IL-1β [251]. Notably, both acute and chronic effects of CUS were abolished in mice lacking the receptor for IL-1 (IL-1R) [251]. In addition to this, Goshen and colleagues [252] showed that increased hippocampal levels of IL-1β are necessary and sufficient to mediate the effects of CMS in mice. Indeed, the deleterious effects of stress on behavior, HPA axis, and neurogenic niche were vanished in IL-1R KO mice, while the chronic brain infusion of IL-1β replicated the behavioral and neurogenic effects of stress. Moreover, whereas adrenalectomy vanished the behavioral effects of stress, chronic treatment with corticosterone in IL-1R KO mice exerted the same behavioral and neurogenic depressant effects as observed in WT control mice. These data strongly suggest that GC release is a downstream mediator of IL-1β, at least in the CMS paradigm. Lastly, in another stress model, intra-hippocampal transplantation of NPCs engineered to overexpress interleukin 1 receptor antagonist (IL-1ra), a physiological IL-1R ligand that does not induce an intracellular response, rescued the number of DCX+ neurons in the DG and the cognitive impairment of SI mice [416].

Collectively these data suggest that IL-1β is the most accountable molecular player involved in stress-induced suppression of neurogenesis.

### 4.6. The Role of Microglia in Stress-Induced Hippocampal Neurogenesis Modulation

#### 4.6.1. Microglia Dysregulation in MDD and Animal Models of Stress

Microglia are the brain-resident innate immune cells with increasingly recognized roles in neuronal function and brain homeostasis, which includes the control of the neurogenic niche in the adult hippocampus [417]. Microglia have been shown to be a relevant cellular component part of the neurogenic niche [418] and to physiologically regulate hippocampal adult neurogenesis at multiple steps of the neurogenesis process, using phagocytosis secretome [419,420], signaling through the CX3C-receptor-1 (CX3CR1) [421,422], and the release of BDNF [423]. In addition, experiments of microglia ablation suggest that microglia are required for basal hippocampal neurogenesis [424]. 

In contrast to neurons, astrocytes, and oligodendrocytes, microglia have a mesodermal derivation, originating from primitive myeloid progenitors during embryonic development [425]. By virtue of such an immunological nature, in physiological conditions, microglia patrol the environment through their thin and elongated processes to easily reply to any noxious insult (pathogens invading the brain, inflammatory stimuli arising from the peripheral blood, pathologically aggregated proteins), playing as antigen-presenting cells or exerting phagocytic activity [426]. Moreover, under transient or pathological circumstances, microglia proliferate and undergo a highly dynamic process of activation that depends on the context and that changes during the pathological process. This explains the high heterogeneity of microglia phenotype in the injured brain, which is reflected by specific transcriptional repertoires ranging from an inflammation-supportive to a reparative one [426,427]. 

Recent studies using single cell flow-cytometry combined with quantitative real-time PCR (qPCR) of MDD postmortem brains have revealed a homeostatic microglia phenotype rather than a proinflammatory state as suggested by histochemical or positron emission tomography (PET) imaging studies [428,429,430]. In addition, PET imaging for translocator protein (TSPO), a marker of microglia, has revealed a significantly attenuated microgliosis in the prefrontal cortex of PTSD patients compared to subjects non-exposed to trauma, and more importantly, a negative correlation between TSPO availability and symptome severity, suggesting brain immune deficiency as the underlying mechanism of PTSD [431]. Such heterogeneity in microglia phenotype highlighted in human studies of both MDD and to a lesser extent PTSD, has also been described in animal models of stress, though most of the data point to a proinflammatory role of microglia [432]. Microglia have been shown to underlie behavioral responses to stress. In a murine model of Gulf War Illness, reduced neurogenesis and principal neuron loss together with mild microgliosis underlined mood and memory deficits [433]. Microglia-depleted mice have been reported to be resistant to develop social avoidance and anxiety-like behavior after exposure to CSDS, and microglial repopulation of the brain post-CSD reintroduced adverse stress effects [434]. Furthermore, consistent with a proinflammatory role of microglia is the finding that microglia of stressed mice express increased surface inflammatory markers and IL-1β [432] and receptor for advanced glycation end products (RAGE), which is involved in inflammasome activation [435]. Nevertheless, a recent paper has clearly shown that two different types of stress (SI vs. repeated injection stress) have divergent effects on HPA axis response, peripheral and central inflammation, as well as microglia activation, hippocampal neurogenesis, and behavioral response [436]. 

#### 4.6.2. Microglia Control of Neurogenesis under Stress Response

Pharmacological targeting of microglia has provided variable results about the contribution of microglia to neurogenesis response under stress. Treatment of mice undergoing repeated social defeat stress with the antibiotic minocycline, known to target microglia, alleviated microglia activation in the hippocampus and spatial memory impairment, but did not rescue impaired neurogenesis or social avoidance [437]. In the restraint stress model, minocycline treatment reduced microglia cell number and hippocampal inflammation and blocked the stress-induced drop of newborn neurons [438]. However, none of these studies addressed the morphological state nor the transcriptional repertoire of microglia, which could have revealed a specific microglia signature linked to neurogenesis in the above stress paradigms. In this respect, compelling evidence comes from a study on CUS [439]. Five weeks of CUS in rats resulted in reduced number, dystrophic morphology, as well as reduced expression of activation markers of microglia in the DG. In contrast, a transient increase of microglia number and activation was observed after an acute CUS exposure (2 days) and followed by microglia apoptosis, suggesting dynamic changes of microglia response during stress. Moreover, minocycline treatment concomitantly with CUS rescued microglia drop and suppressed neurogenesis. Similar results were obtained in mice overexpressing IL-1Ra or treated with the antidepressant imipramine, suggesting that microglia might be a common player of both conditions. Vice versa, treatment aimed to promote microglia proliferation started after CUS induction, increased microglia number, and significantly improved neurogenesis, with minimal antidepressive effects [439]. Overall, these data suggest a dynamic response of microglia to stress that regulates neurogenesis and behavior and depends on several other factors. As recently highlighted by Nieto-Quero and colleagues, microglia activation under stress is a heterogeneous process that depends on the stress characteristics (type and duration), and animal used (age and strain) [440]. 

#### 4.6.3. Targeting Microglia-Specific Signaling Pathways in Animal Stress Models

At the molecular level, research has focused on two surface molecules expressed by microglia, the CX3CR, involved in the fractalkine-mediated microglia-neuron axis, and the purinergic receptor P2X7R. In the brain, the chemokine fractalkine (CX3CL1) is secreted by neurons, and by binding to its receptor CX3CR1 exclusively expressed in microglia, is crucially involved in regulating microglia-mediated synaptic pruning and remodeling, as well as neurogenesis [441]. P2X7R is, a purinergic receptor in the brain that is mainly expressed on microglia and predicted to be involved in the inflammasome pathway leading to the release of proinflammatory cytokines, such as IL-1β [442]. CX3CR1-deficient mice have been found to be resistant to develop depressive-like behavior and microglia hyper-ramification in the hippocampus after exposure to a chronic despair model [443]. More related to the goal of the present review is the finding that CX3CR1 KO mice are resistant to CUS-induced anhedonia and neurogenesis deficits [253]. In particular, the lack of microglial CX3CR1 was shown to impair DG neurogenesis *per se*, without affecting microglia morphology and proliferation, but inducing significant transcriptional changes in the hippocampus, in particular of interferon (IFN) and IFN-related transcripts. Hence, these results indicate that microglial CX3CR1-dependent reduced neurogenesis may be a factor regulating passive mechanisms of resilience to stress. Moreover, these data further highlight the importance of molecular profiling of microglia rather than the mere assessment of density and morphology in addressing the functional role of these cells in the stress response. In contrast to these data, pharmacological targeting of the P2X7R in mice has been proven efficient in reversing the microglia activation, the depressive phenotype, and the HPA axis dysregulation of mice undergoing UCMS with any effects on hippocampal neurogenesis, suggesting a neurogenesis-independent antidepressant activity [444].

Finally, a recent study has highlighted a novel mechanism by which microglia regulate stress resilience-dependent neurogenesis [254]. In a CMS model, the anti-inflammatory marker Arg+, known to be induced by the anti-inflammatory cytokine IL-4, was found up- and downregulated in the microglia of resilient and susceptible mice, respectively, whereas IL-4 appeared to be mainly expressed in neurons of resilient mice. Based on this, the authors addressed the neuron-microglia crosstalk by either knocking-down the IL-4 receptor (IL4R) in microglia or increasing IL-4 signaling in the brain by viral injection, demonstrating the dependence of neurogenesis-driven resilience to stress on IL-4 responsive microglia. Finally, both in vivo and in vitro (cultured NPCs) manipulation of IL-4 signaling show that BDNF is induced by IL-4 and mediates the proneurogenic effects observed in resilient mice.

Overall, these data indicate a dynamic response of microglia during stress that can underlie different behavioral phases and neurogenesis by several mechanisms, linked to both the inflammatory and the BDNF pathways. 

### 4.7. The Role of T Lymphocytes in Stress-Induced Hippocampal Neurogenesis Modulation

MDD patients show peripheral T cell profile alterations that have been claimed to play a role in MDD pathophysiology [445]. Groundbreaking studies have shown that the brain is not an immune-privileged site, and that peripheral T lymphocytes influence brain functioning and behavior in both homeostatic and pathological settings [446]. Different T cell subsets have been implicated in the control of neurogenesis under physiological [447,448,449] and EE conditions [450]. Of particular interest is the finding that immunodeficient mice show impaired hippocampal neurogenesis, whereas transgenic mice engineered to show a circulating pool of T cells recognizing CNS antigens have improved neurogenesis [447]. These data, together with the evidence of T cell suppression in MDD, have led to the notion that T cells could be targeted to improve stress resilience and neurogenesis in MDD. Lewitus and colleagues [255] have shown that rats vaccinated with a peptide with a weak agonism for myelin peptide devoid of encephalitogenic properties were resistant to chronic stress-induced suppression of neurogenesis and to depressive-like behavior, and showed increased hippocampal BDNF levels. Accordingly, T lymphocytes from chronically stressed mice modulated the immune response in recipient mice, skewing microglia profile to an anti-inflammatory state, to confer stress resilience and improving hippocampal neurogenesis, suggesting a potential role of T cells in orchestrating microglia phenotype supporting neurogenesis as a stress resilience mechanism [256]. 

Although few in number, these studies provide strong evidence that boosting T cell function might be a strategy to improve neurogenesis in stress-related disorders.

### 4.8. The Role of Gut-Brain Axis in Stress-Induced Hippocampal Neurogenesis Modulation

The crosstalk between peripheral and central inflammation has increasingly been recognized to occur also through the so-called gut-brain axis, which is dysregulated in patients with MDD [451]. Stress-induced changes in gut microbiota together with leakiness of gut and brain barriers have been associated not only with raises of peripheral and brain cytokines, but also with significant changes of the kynurenine metabolism, thus underlying behavioral, endocrine, and neurogenic outcomes of stress-related disorder (see the updated review on the topic by the authors of [452]). The main reason for targeting the gut-brain axis in the context of psychiatric disorders stems from the fact that 5-HT brain availability depends on proper gut-brain axis functioning, as supported by the following findings: first, for humans, the main source of tryptophan is dietary; second, tryptophan metabolism entirely relies on gut microorganism activity [453]; third, antidepressant drugs depend on 5-HT availability [454]. These are among the main factors that make the gut-brain axis a relevant and novel target for psychiatric disorders, such as MDD and PTSD.

Accumulating evidence indicates that vulnerability to stress is associated with significant gut microbiota alterations [455,456]. Microbiota-depleted mice show resilience in CSDS [457], while dietary supplementation with specific metabolites regulates microbiota composition, as well as gut-associated immune profiling and promotes stress resilience [455,458,459]. Regarding the specific involvement of gut-microbiota in neurogenesis control during stress, combined restraint and CUS stress were shown to induce significant alteration of gut microbiota composition, behavioral and neurogenic impairments [460]. Moreover, preventive dietary interventions based on probiotics have been shown to recover stress-induced raises of plasma corticosterone levels and neurogenesis drop [461]. In contrast, oral supplementation with a specific probiotic formulation has been shown to be ineffective in preventing the depressive-like behavior and the neurogenesis impairment caused by repetitive corticosterone-injections, but improved HPA axis response [462]. Such discrepancies may arise from the different strains of probiotics used in these studies. However, it is noticeable that both treatments had some impact on the HPA axis.

Noteworthy, the causal relationship between gut microbiota dysfunction and impaired neurogenesis after the stress has been convincingly provided by Siopi and colleagues [257]. The transfer of gut microbiota from stressed mice into healthy recipient mice was shown to induce depressive-like behavior, impair hippocampal neurogenesis, and significantly affect tryptophan metabolism. Indeed, serum metabolomic analysis showed reduced levels of the 5-HT precursor 5-hydroxytryptophan (5-HTP), which was reflected by low hippocampal 5-HT levels. Notably, fluoxetine treatment in microbiome transplanted mice did not restore brain 5-HT levels and did not correct behavioral and neurogenesis alterations, while 5-HTP administration fully rescued behavioral and neurogenic functions [257]. Although the microorganisms involved in kynurenine metabolism have not been identified, these data clearly demonstrate that stress induces significant changes in gut microbiota function, which in turn alters 5-HT availability, thus impairing fluoxetine efficacy. 

Beyond the control of the kynurenine pathway, the gut-brain axis might influence hippocampal neurogenesis during stress also by regulating both the HPA axis and immune system [463]. This issue is worth to be investigated.

## 5. Resilience: Is Neurogenesis a Resilience Mechanism?

As outlined above, at least in rodent models, long-term treatment with GCs and chronic stress reduces adult hippocampal neurogenesis. Furthermore, besides the neurogenesis-independent action of antidepressants, it is well established that antidepressants restore adult neurogenesis, which is necessary for their antidepressant effect on behavior [116,117]. Nevertheless, it remains elusive whether a reduction of neurogenesis directly causes depression-like behavior or whether this is an epiphenomenon. Several studies reported that ablating adult neurogenesis in rodents is sufficient to induce an anxiety-/depressive-like phenotype in the forced swim or tail suspension test, and in the novelty-suppressed feeding paradigm [38,39,40]. However, other studies with stress-induced impairment in neurogenesis report divergent results [77,79,464,465], supporting the notion that decreased adult neurogenesis is rather a risk factor than an actual cause for MDD development. 

Stress resilience is defined as the absence of mental disease despite adversity. It is commonly accepted that there exists a predisposition of an individual to allow maintenance or rapid adaptation and recovery of mental health during and after periods of stressor exposure. This individual predisposition is supposed to arise from individual traits or characteristic properties, either gained by genetic predisposition and/or environmental factors [466,467]. We are far from understanding resilience mechanisms, even if some factors or conditions, e.g., BDNF and PE, have been described to act in a beneficial or preventive manner against stress-related symptoms. One reason is that the concept of individual analysis has just recently emerged. One further critical issue is the time point, when behavioral, neurophysiological, or molecular parameters are measured, i.e., before stress, during stress, or after stress. Proper results ideally would imply multiple data acquisitions. In this respect, the longitudinal tracking of an animal or human individual is central, which can be at least performed on the behavioral level or immune system level. In this respect, a study from Hodes and colleagues has shown that pre-existing individual differences in the peripheral immune system predict the susceptibility or the resilience to stress in CSDS [139]. Of course, the analysis of blood-derived factors easily allows for multiple sampling to draw trajectories of resilience or susceptibility in association with other non-invasive measures [468]. However, when testing CNS correlates, multiple time points of acquisition are difficult to achieve. This applies particularly to adult neurogenesis, which so far cannot be specifically measured in vivo, e.g., by MRI methods [469,470]. The lack of sensitivity by these methods does not exactly measure the amount of new-built granule cells in contrast to measuring hippocampal volume, which, however, is feasible in animals and humans. For this reason, it is currently only possible to find poststress correlates of neurogenesis in animals, which mainly describe the adaptation process; but the resilient outcome could also be due to distinct individual levels of neurogenesis before stress, constituting an individual trait. When manipulating neurogenesis before stress, e.g., by ablation or drug-/running-dependent enhancement, only the time point before stress is considered, which would mainly suggest neurogenesis as a predisposing factor. Besides this, individual resilience factors or traits, which maintain the equilibrium, meaning keeping the “optimal” level of neurogenesis constant, during stress might be important in classifying neurogenesis as a resilience mechanism (discussed by the authors of [471]). 

### 5.1. Analysis of Adult Neurogenesis after Stress

A study by Jayatissa and colleagues resolved the temporal issue of stress-related behavior and neurogenesis by analyzing the time course of events after stress exposure. The authors demonstrated that in CMS-stressed animals, anhedonic symptoms appeared prior to a decrease of neurogenesis. Moreover, there was no correlation between deficits in neurogenesis and anhedonic behavior when analyzing neurogenesis rates after stress in the susceptible and resilient groups [464]. This speaks against a direct disease-promoting role of reduced neurogenesis and agrees with experiments using LH-stressed rats. Moreover, using this paradigm, diminished cell proliferation appeared after symptoms of helplessness without correlating to the individual resilient or susceptible behavioral phenotype [465]. Unfortunately, only a few assays to test depressive-like behavior were performed (sucrose preference, helplessness paradigm), lacking other important assays, such as the forced swim test. It can be concluded that regulating adult neurogenesis by stress is not the only factor leading to the emergence of a depressive-like phenotype in rodents. Although this does not exclude the possibility that a stress-induced reduction of adult neurogenesis can result in the development of MDD when predisposing, e.g., genetic factors or other negative environmental conditions are present. For this reason, it is conceivable that a constitutive low level of hippocampal neurogenesis, due to long-lasting chronic stress, can be an important factor that predisposes an individual to the emergence of depression. 

The study from Lagace and colleagues (2010) brought up another interesting aspect. The authors demonstrated that CSDS-induced changes, despite a behaviorally-independent overall decrease of proliferation, led only in susceptible animals to increased survival of newly generated neurons born 24 h after stress [77]. These data suggest a stress-induced compensatory enhancement of adult neurogenesis, which, however, seems to lead to long-term individual maladaptive responses to stress, as susceptible animals displayed persistent social avoidance. 

### 5.2. Manipulation of Adult Neurogenesis before Stress

All results mentioned above have the shortcoming that they show stress effects on adult neurogenesis after the stress procedure, and therefore, solely address the role of stress-induced changes on neurogenesis and how this relates to a depressive-like phenotype. However, an important remaining question is whether an increased or decreased neurogenesis before stress protects from or renders vulnerable to develop MDD symptoms. Interestingly, in the study by Lagace et al. (2010), when animals were subjected to irradiation to ablate neurogenesis four weeks before CSDS, the percentage of susceptible animals was attenuated, which accounts for a negative role of adult neurogenesis for resilient behavior. Another study using CMS described no negative effect of neurogenesis on resilient behavior. Here, irradiation-ablated neurogenesis before stress did not aggravate sensitivity to CMS as tested in the novelty-suppressed feeding and the grooming behavior-analyzing splash test [472]. In contrast, experiments with chronic restraint stress demonstrated that adult neurogenesis is necessary to buffer HPA axis-controlled stress responses and anxiety/depressive-like behavior. Neurogenesis-ablated animals displayed a slower recovery of GC levels after moderate stress and less dexamethasone-induced suppression of GC levels, which was manifested behaviorally in increased anhedonia, reduced latency to immobility in the forced swim test, and anxiety-like behavior in the novelty-suppressed feeding paradigm [38]. In line with these data are studies using iBax mice crossbred with Nestin-CreERT2 mice, in which an inducible stem-cell specific KO of the proapoptotic protein bax leads to increased neurogenesis. It was shown that increased neurogenesis before chronic GC treatment or UCMS promoted resilience by reducing anxiety and depressive-like symptoms [41,473]. In a similar study with UCM-stressed iBax mice, not all behavioral symptoms, but anhedonia was attenuated. Importantly, the increase of adult neurogenesis was sufficient to reverse HPA axis deficits [42]. In reverse, one could imagine that individuals with baseline reduction of adult neurogenesis, hence dysregulated HPA axis, and therefore, impaired control of stress, could be more susceptible to future stressful circumstances and harbor reduced ability to cope with them in an adequate manner. Importantly, Anacker and colleagues recently reported that elevated neurogenesis rates in iBax mice led to decreases in the activity of ventral stress-responsive mature granule cells, which was sufficient to confer resilience to CSDS, whereas ablating ventral neurogenesis by irradiation led to susceptibility [45]. 

### 5.3. Conclusions

Altogether, the current data point towards a disease-preventing role of adult neurogenesis, which, however, might not be sufficient to produce complete protection against all stress-induced behavioral impairments. The above mentioned contradicting results could have been due to experimental reasons by using different stress paradigms and neurogenesis ablation protocols, as well as distinct behavioral readouts. Age, species (mice, rats), and strain of animals could further play a role. Translating a protecting role of neurogenesis to humans is still ambiguous and could be rather based on indirect results obtained in humans, which can be related to rodent studies. For example, it is known that PE in humans, especially running, acts as an antidepressant intervention by increasing hippocampal volumes and also reversing the age-dependent volume decline [204,474]. In addition, it is commonly accepted that PE contributes to protection against psychiatric disorders [475]. As mentioned above, in rodents, PE strongly increases neurogenesis [150], but this direct link is missing in humans, and therefore, interpretation can be only indirect. Observed changes in human hippocampal volumes could also be attributed to enhanced synaptic plasticity by neurite extensions or changes in dendritic or spine morphology. Altogether, even if very likely, direct evidence of whether and how adult neurogenesis modulates stress resilience in humans is missing.

However, as revealed in rodent studies, there is strong evidence that adaptive capacities to stressors are supported by adult hippocampal neurogenesis [471]. Particularly, the mentioned studies modifying adult neurogenesis before stressor exposure, propose that neurogenesis is one of the predisposing factors being beneficial for appropriate stress coping. As it has already been demonstrated by Freund and colleagues, the emergence of individual exploratory behavior positively correlates with individual changes in neurogenesis [54]. Moreover, a recent study by Milic and colleagues demonstrated that CSDS-resilient animals displayed higher exploratory drive to a novel environment, but also to social and non-social targets, whereas susceptible mice were better in learning the passive avoidance task, which suppressed their spontaneous exploratory drive [476]. This means that individual baseline behavior can predict resilience or susceptibility to stress. Taking together both publications, it can be suggested that more individual exploratory drive, hence “resilience”, equals higher individual rates of neurogenesis, which finally suggests adult hippocampal neurogenesis as one of many resilience factors, if not a mechanism. However, further research is needed to clarify to which extent and in which connection it stands to other predisposing factors.

## Figures and Tables

**Figure 1 ijms-22-07339-f001:**
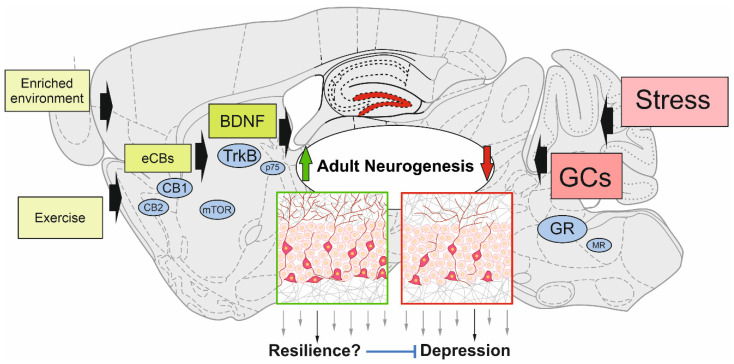
Schematic view of modulating factors of adult hippocampal neurogenesis. Enriched environment, exercise, and molecular players (e.g., endocannabinoids (eCBs) and brain-derived neurotrophic factor (BDNF)) have the potential to upregulate the generation of adult-born neurons in the dentate gyrus. This could confer resilience to the development of depressive-like symptoms through the stress-related decline of adult neurogenesis induced by glucocorticoids (GCs). The main signaling pathways of positive modulators and stress are depicted: cannabinoid receptor type-1 and -2 (CB1; CB2); mammalian target of rapamycin (mTOR); tropomyosin receptor kinase B (TrkB); p75 neurotrophin receptor (p75); glucocorticoid receptor (GR); mineralocorticoid receptor (MR).

**Figure 2 ijms-22-07339-f002:**
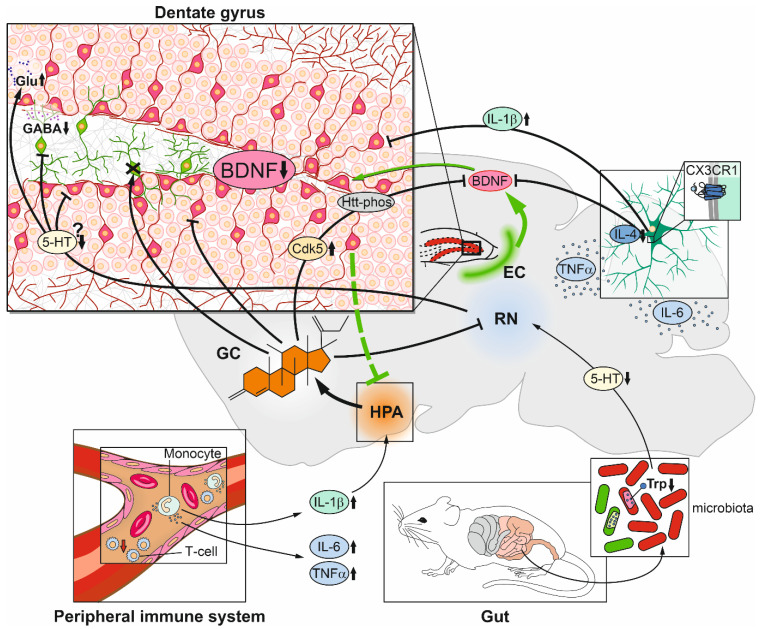
Schematic integrated view of mechanisms conferring negative regulation of adult hippocampal neurogenesis by chronic stress. Hypothalamic–pituitary–adrenal-(HPA) axis controls stress-induced glucocorticoid (GC) release, which exerts a direct negative effect on adult neural stem cells and their progeny (red cells) in the dentate gyrus. In reverse, adult-born neurons exert negative feedback on the HPA axis (green dashed line). HPA axis can get activated by increased interleukin-1β(IL-1β) released by peripheral monocytes. Under conditions of chronic stress, the gut microbiome can change to low tryptophan (Trp) metabolizing microbiota, mainly targeting the serotonergic system of the raphe nuclei (RN), leading to serotonin (5-HT) reduction. 5-HT-reduction might also be due to direct GC effects on raphe neurons, with control of hippocampal glutamatergic and GABAergic release in the dentate gyrus, leading to high glutamate (Glu) and reduced GABA levels. Low GABA is further caused by a GC-mediated decrease of hilar interneurons (green cells). In addition, reduced 5-HT and dysregulation of 5-HT receptors (5HT_1A_R and 5HT_4_R, not depicted) on mature granule cells (light red cells) leading to reduced BDNF availability might be implicated in the stress-induced reduction of neurogenesis. BDNF reduction is also central, due to other mechanisms: (1) Decreased axonal transport of BDNF vesicles from the entorhinal cortex (EC; green arrows for vesicular BDNF-transport), due to hyperphosphorylated huntingtin (Htt-phos) mediated by GC-induced cyclin-dependent kinase 5 (Cdk5); (2) decrease in interleukin-4 (IL-4) sensed by microglia. Moreover, microglia directly inhibit adult neural stem cells and their progeny, (1) through signaling activated by binding of the chemokine CX3CL1 to CX3CR1, which is exclusively expressed on microglia, and (2) by the secretion of IL-1β. Increased interleukin-6 (IL-6) and tumor necrosis factor-α (TNFα) make up the inflammatory milieu of the brain, but are not necessarily released by microglia.

**Table 1 ijms-22-07339-t001:** Summary of different chronic stress protocols in rodents, their behavioral outcome, and effect on adult hippocampal neurogenesis.

Protocol of Stress	Behavior	Effect on Neurogenesis (↓ Decreased; ↔ Unchanged; ↑ Increased)
Chronic social stress, Chronic social defeat stress (CSDS)	↑ Anhedonia, Social avoidance, ↑ Sleep disturbances, ↓ Exploratory anxiety, ↓ Weight	↓ Simon et al., 2005; Schloesser et al., 2010; Jiang et al., 2019 [73,74,75] ↔ Hanson et al., 2011 [76] ↑ Lagace et al., 2010 [77]
Chronic unpredictable stress (CUS), (Unpredictable) Chronic mild stress ((U)CMS)	↑ Anhedonia, ↑ Sleep disturbances, ↑ Behavioral despair, ↓ Grooming, ↓ Weight	↓ Jayatissa et al., 2006, 2009; Toth et al., 2008; Surget et al., 2011; Dioli et al., 2017 [78,79,80,81,82] ↔ Lee et al., 2006 [83]
Chronic corticosteroid treatment	↑ Anhedonia, ↑ Behavioral despair, ↑ Anxiety	↓ Ekstrand et al., 2008; Brummelte and Galea, 2010; Pazini et al., 2017; Levone et al., 2020 [84,85,86,87]
Repeated restraint stress	↑ Anhedonia, ↑ Anxiety, ↑ Behavioral despair	↓ Luo et al., 2005; Rosenbrock et al., 2005; Snyder et al., 2011 [38,88,89] ↔ O’Leary et al., 2012 [90] ↑ Parihar et al., 2011 [91]
Early life stress (ELS)	↑ Anhedonia, ↑ Anxiety, ↑ Behavioral despair, ↓ Learning, ↓ Locomotion	↓ Mirescu et al., 2004; Kikusui et al., 2009; Lajud et al., 2012 [92,93,94]
Prenatal (restraint of pregnant dams)	↑ Anhedonia, ↑ Anxiety, ↑ Behavioral despair	↓ Lemaire et al., 2000; Bosch et al., 2006; Mandyam et al., 2008; Lucassen et al., 2009 [95,96,97,98]
Learned helplessness (chronic tail or footshocks) (LH)	↓ Active avoidance, ↑ Sleep disturbances, ↓ Weight	↓ Malberg and Duman, 2003 [99] ↔ Van der Borght et al., 2005 [100]
Social isolation (SI)	↑ Anxiety, ↑ Behavioral despair, ↓ Learning,	↓ Westenbroek et al., 2004; Spritzer et al., 2011; Chan et al., 2017 [101,102,103]
Lipopolysaccharide-induced sickness behavior	↑ Anhedonia, ↑ Lethargy, ↓ Appetite and food intake, ↑ Anxiety	↓ Ekdahl et al., 2003; Monje, 2003; Yirmiya and Goshen, 2011; Perez-Dominguez et al., 2019 [104,105,106,107] ↔ Depino, 2015 [108]

**Table 2 ijms-22-07339-t002:** Summary of key publications describing diverse mechanisms leading to stress-reduced adult hippocampal neurogenesis.

**HPA Axis (Section 4.1)**				
**Stress/Rodent or Cellular Model**	**Proposed Mechanism**	**Output on Adult Neurogenesis** **(↓ Decreased; ↔ Unchanged; ↑ Increased)**	**Susceptibility (S)/Resilience (R) factor?**	**Reference**
Corticosterone treatment of human hippocampal progenitor cell line in the presence of: SGK1 antagonist GR antagonist	SGK1 mediates the effects of cortisol on neurogenesis by inhibiting the Sonic hedgehog pathway and by inhibiting GR phosphorylation and nuclear translocation	corticosterone-treated cells in the presence of SGK1 antagonist/GR antagonist ↓ BrdU+ ↓ DCX+	SGK1 (S)	Anacker et al., 2013 [245]
**BDNF decrease (Section 4.2)**				
**Stress/rodent or cellular model**	**Proposed mechanism**	**Output on adult neurogenesis**	**Susceptibility (S)/Resilience (R) factor?**	**Reference**
Chronic corticosterone treatment in wild-type, and unphosphorylatable htt mutant mice (Hdh ^S1181A/S1201A^)	Cdk5-mediated hyperphosphorylation of htt impairs BDNF transport to the DG	in wild-type, but not in mutant mice: ↓ Ki67+, proliferation ↓ BrdU+, survival ↓ BrdU+/calbindin+, maturation (↔ DCX+, immature neurons)	Cdk5 (S) Htt (R?) BDNF (R)	Agasse et al., 2020 [246]
Chronic unpredictable stress in wild-type and tau KO mice	Chronic stress triggers tau hyperphosphorylation and alters tau isoforms, reduced PI3K/mTOR/GSK3β/β-catenin pathway	in wild-type, but not in KO mice: ↓ BrdU+, survival, and proliferation ↓ Ki67+, proliferation ↓ BrdU+/Ki67+, proliferation ↓ BrdU+/DCX+, differentiation	Tau (S?)	Dioli et al., 2017 [82]
**5-HT signaling (Section 4.3)**				
**Stress/rodent or cellular model**	**Proposed mechanism**	**Output on adult neurogenesis**	**Susceptibility (S)/Resilience (R) factor?**	**Reference**
Chronic SSRI treatment of mice lacking 5-HT_1A_R either on mature (floxed-5-HT_1A_R x POMC-Cre mice) or young adult-born DG (floxed-5HT_1A_R x Nestin-CreERT2 mice) granule cells; acute inescapable stress in FST, NSF	5-HT_1A_R on mature, but not on young adult-born granule cells is sufficient for the SSRI effect, due to BDNF expression in mature granule cells	in floxed-5-HT_1A_R x POMC, but not in floxed- 5-HT_1A_R x Nestin-CreERT2 mice: ↓ BrdU+, proliferation ↓ DCX+, differentiation ↓ DCX+ with tertiary dendrites, maturation	5-HT_1A_R (R?)	Samuels et al., 2015 [247]
Chronic SSRI treatment in wild-type and 5-HT_4_R KO mice	5-HT_4_R mediated SSRI-effect on neurogenesis correlates with BDNF-mediated dematuration of DG cells	in 5-HT_4_R KO, but not in wild-type mice: ↓ BrdU+, proliferation ↓ DCX+, differentiation ↓ Calbindin+, dematuration of DG cells	5-HT_4_R (R?)	Imoto et al., 2015 [248]
**Excitation/inhibition imbalance (Section 4.4)**				
**Stress/rodent or cellular model**	**Proposed mechanism**	**Output on adult neurogenesis**	**Susceptibility (S)/Resilience (R) factor?**	**Reference**
Social isolation stress of PV-Cre mice DG-injected with AAV-DIO-ChR2 with or without optogenetic activation	PV+ interneurons activation restores quiescence of RGLs	in AAV-DIO-ChR2 PV-Cre mice without photoactivation, but not in photoactivated: ↑ EdU+/Nestin+, proliferation of quiescent pool ↑ MCM2+/Nestin+, activation of quiescence	PV+ interneurons (R?)	Song et al., 2012 [249]
Deletion of γ2GABA_A_R subunit in immature neurons of embryonic and adult forebrain (Emx1-Cre x γ2+) or mature neurons in adulthood (CaMKII-Cre2834 x γ2+); acute inescapable stress in FST	A developmental, but not adult deficit of γ2GABA_A_R subunit leads to depressive-like traits in adults	in Emx1Cre x γ2+, but not in CaMKIICre2834 x γ2+ mice: ↓ BrdU+/NeuN+, differentiation (↔ BrdU+, proliferation)	γ2GABA_A_R subunit (R)	Earnheart et al., 2007 [250]
**Cytokines (Section 4.5)**				
**Stress/rodent or cellular model**	**Proposed mechanism**	**Output on adult neurogenesis**	**Susceptibility (S)/Resilience (R) factor?**	**Reference**
Intracerebroventricular infusion of IL-1β in control mice; Intracereroventricular infusion of IL-1R antagonist in mice undergoing chronic unpredictable stress; chronic unpredictable stress in IL-1R KO mice	Il1-β-dependent activation of nuclear factor kB	in control mice receiving IL-1β: ↓ BrdU+ in stressed IL-1R KO mice or stressed mice receiving intracerebroventricular IL-1R antagonist: ↑ BrdU+ ↑ DCX+	IL-1β brain expression (S)	Wook Koo and Duman, 2008 [251]
CMS in IL-1R KO mice Corticosterone treatment in IL-1R KO mice Intracerebroventricular infusion of IL-1β in control mice;	Corticosterone is a downstream mediator of IL-1β	in IL-1R KO stressed mice: ↔ BrdU+ ↔ DCX+ in IL-1R KO mice treated with corticosterone: ↓ DCX+ ↓ Ki67+ in control mice receiving IL-1β: ↓ BrdU+ ↓ DCX+	IL-1β brain expression (S)	Goshen et al., 2008 [252]
**Microglia (Section 4.6)**				
**Stress/rodent or cellular model**	**Proposed mechanism**	**Output on adult neurogenesis**	**Susceptibility (S)/Resilience (R) factor?**	**Reference**
Chronic unpredictable stress in CX3CR1 KO mice	In CX3CR1 KO mice in basal condition, a reduced transcription of MHC-I and downstreamof IFNs and altered transcription downstream of 17-β-estradiol. After stress, CX3CR1 KO show no reductions in transcriptional regulation downstream of ESR2	in CX3CR1 KO mice in basal condition: ↓ DCX+ in stressed CX3CR1 KO mice: ↔ DCX+	CX3CR1 (R)	Rimmerman et al., 2017 [253]
Chronic mild stress in microglial IL-4R KO mice (lentivirus vector with LoxP-shIL4Rα injected into the hippocampus of CX3CR1-CreERT2) injected with AAV or AAV-IL-4 Chronic mild stress in C57BL6 mice receiving AAV-IL-4 hippocampal injection treated or not with TrkB antagonist	IL4-responsive microglia regulates BDNF	in chronic mild stress in microglial IL4-R KO mice (lentivirus vector with LoxP-shIL4Rα injected into the hippocampus of CX3CR1Cre/ERT2) injected with AAV or AAV-IL-4: proliferating and differentiating cells (BrdU given after stress) ↔ BrdU+ ↓ BrdU+/DCX+ surviving proliferative cell (BrdU given before stress) ↓ BrdU+ ↓ BrdU+/NeuN+ in C57BL6 stressed mice receiving AAV-IL-4 hippocampal injection treated with TrKB antagonist: ↓ BrdU+/DCX+ ↓ DCX+	IL-4 brain expression (R)	Zhang et al., 2021 [254]
**T lymphocytes (Section 4.7)**				
**Stress/rodent or cellular model**	**Proposed mechanism**	**Output on adult neurogenesis**	**Susceptibility (S)/Resilience (R) factor?**	**Reference**
Chronic mild stress in rats treated with A91, a modified peptide cross-reacting with the original MBP-derived peptide	Induction of neuroprotective mechanisms through BDNF signaling	in stressed mice treated with A91: ↑ BrdU+ ↑ BrdU+/DCX+	T cell immune responsiveness (R?)	Lewitus et al., 2009 [255]
Transfer of T cells from stressed mice into recipient Rag2-/- mice Chronic social defeat stress model	Induction of peripheral anti-inflammatory effects and microglia supporting neuroprotective effects	in recipient Rag2-/- mice receiving T cells of stressed donors: ↑ BrdU+	T cells retaining memory of stress experiences (R?)	Brachman et al., 2015 [256]
**Gut-brain axis (Section 4.8)**				
**Stress/rodent or cellular model**	**Proposed mechanism**	**Output on adult neurogenesis**	**Susceptibility (S)/Resilience (R) factor?**	**Reference**
Transfer of gut microbiota from stressed donor mice treated or not with fluoxetine into antibiotics-treated recipient mice; Transfer of gut microbiota from stressed donor mice treated or not with fluoxetine into antibiotics-treated recipient mice supplemented with tryptophan; Unpredictable chronic mild stress	Gut microbiota-dependent tryptophan metabolism restores serotonin levels necessary for fluoxetine antidepressant and neurogenic effects	in recipient mice receiving gut microbiota from stressed donor mice treated or not with fluoxetine: ↓ DCX+ ↓ Ki67+ in recipient mice receiving gut microbiota from stressed donor mice treated or not with fluoxetine, supplementation with tryptophan: ↑ DCX+ ↑ Ki67+	Gut tryptophan (R)	Siopi et al., 2020 [257]

## Data Availability

Not applicable.

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
