# Peer review of "Stress-Related Dysfunction of Adult Hippocampal Neurogenesis—An Attempt for Understanding Resilience?"

_ijms, 2021, doi:10.3390/ijms22147339_

Round 1

Reviewer 1 Report

The manuscript entitled “Stress-Related Dysfunction of Adult Neurogenesis—An Attempt for Understanding Resilience? is a good review of the role that plays neurogenesis in resilience to the negative effects of stress. Although there are other reviews on the subject it approaches it from a different and updated perspective on this topic. The paper is well written with an extensive literature review.

I suggest some minor changes:

1.- In the introduction too much is devoted to neurogenesis in the subventricular zone, but the article focuses on hippocampal neurogenesis. This part should be shortened a little and "hippocampal neurogenesis" should be included in the title.

2.- The following sentence can be misleading. It is not resilience that is related to the development of depression and suicidal ideation. Please rephrase:  “In a recent study, Boldrini et al. (2019) reported that resilience to early life adversity, which is a risk factor to develop MDD, including suicidal behavior in adulthood [, is associated with an increased DG, presumably due to increased neurogenesis during childhood

3 Figure 1: Schematic view of modulating factors of adult hippocampal neurogénesis could be further enriched by elaborating on the pathways involved in the modulation of neurogenesis.

4.- On line 324-325 authors the authors allude to the mTOR pathway. They should elaborate a little more on why normalisation of this pathway is important for the prevention of CUS-induced impaired adult neurogenesis in the SGZ induced by CUMS.

On line 980-981 authors indicated that there is a time-dependent dynamic response of microglia to stress that regulate neurogenesis and behavior that may explain the atrophy of microglia after 5 weeks of stress, however there are other factors than may be involved. For a more detailed description please see the recent systematic review entitled “Do changes in microglial status underlie neurogenesis impairments and depressive-like behaviours induced by psychological stress? A systematic review in animal models, Neurobiology of Stress, 2021, 100356, https://doi.org/10.1016/j.ynstr.2021.100356.

5.- On line 931-933. Authors refer to the passage of microglia from a resting state to different states of activation. However, microglia are always active in a "surveillance" state, so this expression should be modified. Moreover, authors indicated that “to several shades of activation (of microglia) that have been generally divided into two distinct categories, e.g., M1 pro-inflammatory and M2 anti- inflammatory”. The M1 vs. M2 concept is now considered to be outdated and not thought to be very relevant for microglia in vivo, as most microglia express a mix of markers thought to be representative of each state (which was categorized in vitro). these terms should be removed and replaced by more up-to-date expressions and references.

Author Response

Please see in attachment.

Reviewer 2 Report

Leschik et al. discuss the relation between adult neurogenesis, glucocorticoid exposure and stress resilience. I found it a nice, extensive and quite complete overview of this field by well known scienctists. I have only very few to add except a few comments on additional suggestions to incorporate.

While they provide a nice literature overview, the main part of their aim and title focuses on whether or not NG contributes to actual resilience. This part could be expanded somewhat more, as I feel it still remains somewhat superficial and open now.

In general, for their layout, I would suggest to use more paragraphs, e.g. in their introduction, and then split the text on SVZ / SGZ, on the hippocampus and on depression/PTSD, etc. and also consider using more subheadings to better (re-)organize aspects that belong together. The current ‘monologue’ of a large piece of text will then be easier to read. 
Also, the authors could shorten parts by e.g. reducing text on e.g. the biochemistry of depression, 5HT, NMDA-R, IDO-1, and in general the ‘textbook’ parts, but try to focus in as much as the topic relates to neurogenesis. There appears to be an empty page between 17 and 18.

Next to modulation of neurogenesis by transgenic means, effects of environmental factors like (postnatal) early life stress and its mediators and how this can 'program' neurogenesis for life (TINS), as well as stimuli like ECT and epilepsy they also affect adult neurogenesis and stress levels, with considerable relevance for this paper, which are now discussed only to a limited extent. Some more relevant papers on this could be briefly discussed; Naninck et al., Hip2015; Nuninga et al., MP 2020; Jonckheere et al., 2018; Schloesser 2015; Gbyl PNBP21, etc . 

For the debate on human neurogenesis, 2 papers by Lucassen his group critically summarize key elements that help understand some of the causes of the controversy and deserve to be cited; Lucassen et al., Mol Psy 2020 and BehavBrainRes 2021.

For their remark on adrenalectomy on p.12, l 548, it will be important to highlight that this treatment also induces extensive apoptosis selectively in the DG (Sloviter 89), which, given the tight, time-dependent relation of cell death with neurogenesis (Dupret/Abrous Plos Biol 2007), will be important to highlight.

The same applies to the functional role of neurogenesis in cognition, for which the different roles of the different ages of the newborn cells has recently been discussed in NatCom and MP (Abrous group).

Other than that, no main comments; very nice overview!

Naninck EF, Hoeijmakers L, Kakava-Georgiadou N, Meesters A, Lazic SE, Lucassen PJ, Korosi A. Chronic early life stress alters developmental and adult neurogenesis and impairs cognitive function in mice. Hippocampus. 2015 Mar;25(3):309-28. doi: 10.1002/hipo.22374. Epub 2014 Oct 30. PMID: 25269685

A Baldwin interpretation of adult hippocampal neurogenesis: from functional relevance to physiopathology.
Abrous DN, Koehl M, Lemoine M.Mol Psychiatry. 2021 Jun 8. doi: 10.1038/s41380-021-01172-4. Online ahead of print.PMID: 34103674 Review.

Adult-born neurons immature during learning are necessary for remote memory reconsolidation in rats.
Lods M, Pacary E, Mazier W, Farrugia F, Mortessagne P, Masachs N, Charrier V, Massa F, Cota D, Ferreira G, Abrous DN, Tronel S.Nat Commun. 2021 Mar 19;12(1):1778. doi: 10.1038/s41467-021-22069-4. 

Spatial learning depends on both the addition and removal of new hippocampal neurons.
Dupret D, Fabre A, Döbrössy MD, Panatier A, Rodríguez JJ, Lamarque S, Lemaire V, Oliet SH, Piazza PV, Abrous DN.PLoS Biol. 2007 Aug;5(8):e214. 

Selective loss of hippocampal granule cells in the mature rat brain after adrenalectomy.
Sloviter RS, Valiquette G, Abrams GM, Ronk EC, Sollas AL, Paul LA, Neubort S.Science. 1989 Jan 27;243(4890):535-8. doi: 10.1126/science.2911756

Gbyl K, Rostrup E, Raghava JM, Andersen C, Rosenberg R, Larsson HBW, Videbech P. Volume of hippocampal subregions and clinical improvement following electroconvulsive therapy in patients with depression. Prog Neuropsychopharmacol Biol Psychiatry. 2021 Jan 10;104:110048. doi: 10.1016/j.pnpbp.2020.110048. Epub
2020 Jul 28. PMID: 32730916.

Nuninga JO, Mandl RCW, Boks MP, Bakker S, Somers M, Heringa SM, Nieuwdorp W, Hoogduin H, Kahn RS, Luijten P, Sommer IEC. Volume increase in the dentate gyrus after electroconvulsive therapy in depressed patients as measured with 7T. Mol
Psychiatry. 2020 Jul;25(7):1559-1568. doi: 10.1038/s41380-019-0392-6. Epub 2019
Mar 12. PMID: 30867562.

Jonckheere J, Deloulme JC, Dall'Igna G, Chauliac N, Pelluet A, Nguon AS, Lentini C, Brocard J, Denarier E, Brugière S, Couté Y, Heinrich C, Porcher C, Holtzmann J, Andrieux A, Suaud-Chagny MF, Gory-Fauré S. Short- and long-term efficacy of electroconvulsive stimulation in animal models of depression: The essential role of neuronal survival. Brain Stimul. 2018 Nov-Dec;11(6):1336-1347.
doi: 10.1016/j.brs.2018.08.001. Epub 2018 Aug 15. PMID: 30146428.

Schloesser RJ, Orvoen S, Jimenez DV, Hardy NF, Maynard KR, Sukumar M, Manji HK, Gardier AM, David DJ, Martinowich K. Antidepressant-like Effects of Electroconvulsive Seizures Require Adult Neurogenesis in a Neuroendocrine Model of Depression. Brain Stimul. 2015 Sep-Oct;8(5):862-7. doi:
10.1016/j.brs.2015.05.011. Epub 2015 Jun 9. PMID: 26138027; PMCID: PMC4567930.

Lucassen PJ, Fitzsimons CP, Salta E, Maletic-Savatic M. Adult neurogenesis, human after all (again): Classic, optimized, and future approaches. Behav Brain Res. 2020 Mar 2;381:112458. doi: 10.1016/j.bbr.2019.112458. Epub 2019 Dec 30.
PMID: 31899214.

Lucassen PJ, Toni N, Kempermann G, Frisen J, Gage FH, Swaab DF. Limits to human neurogenesis-really? Mol Psychiatry. 2020 Oct;25(10):2207-2209. doi:
10.1038/s41380-018-0337-5. Epub 2019 Jan 7. PMID: 30617274; PMCID: PMC7515796.

Joshi SH, Espinoza RT, Pirnia T, Shi J, Wang Y, Ayers B, Leaver A, Woods RP, Narr KL. Structural Plasticity of the Hippocampus and Amygdala Induced by Electroconvulsive Therapy in Major Depression. Biol Psychiatry. 2016 Feb
15;79(4):282-92. doi: 10.1016/j.biopsych.2015.02.029. Epub 2015 Mar 5. PMID: 25842202; PMCID: PMC4561035.

Lucassen PJ, Naninck EF, van Goudoever JB, Fitzsimons C, Joels M, Korosi A. Perinatal programming of adult hippocampal structure and function; emerging roles of stress, nutrition and epigenetics. Trends Neurosci. 2013 Nov;36(11):621-31. doi: 10.1016/j.tins.2013.08.002. Epub 2013 Aug 30. PMID: 23998452.

Author Response

Please see in attachment.
